# Technical note: Mapping surface saturation dynamics with thermal infrared imagery

Barbara Glaser[1,2], Marta Antonelli[1,3], Marco Chini[4], Laurent Pfister[1,5], Julian Klaus[1]

[1]Catchment and Eco-Hydrology Research Group, Luxembourg Institute of Science and Technology, Esch/Alzette, 4362, Luxembourg
[2]Department of Hydrology, University of Bayreuth, Bayreuth, 95447, Germany
[3]Hydrology and Quantitative Water Management Group, Wageningen University & Research, Wageningen, 6700, The Netherlands
[4]Remote Sensing and Ecohydrological Modelling Research Group, Luxembourg Institute of Science and Technology, Esch/Alzette, 4362, Luxembourg
[5]Faculty of Science, Communication and Technology, University of Luxembourg, Maison du Savoir, 2 Avenue de l'Université, L-4365 Esch-sur-Alzette

*Correspondence to*: Barbara Glaser (barbara.glaser@list.lu)

## Abstract

Surface saturation can have a critical impact on runoff generation and water quality. Saturation patterns are dynamic and thus their potential control on discharge and water quality is also variable in time. In this study, we assess the practicability of applying thermal infrared (TIR) imagery for mapping surface saturation dynamics. The advantages of TIR imagery compared to other surface saturation mapping methods are its large spatial and temporal flexibility, its non-invasive character, and the fact that it allows a rapid and intuitive visualization of surface-saturated areas. Based on an 18-month field campaign, we review and discuss the methodological principles, in which conditions the method works best, and the problems that may occur. These considerations enable potential users to plan efficient TIR imagery mapping campaigns and benefit from the full potential offered by TIR imagery, which we demonstrate with several application examples. In addition, we elaborate on image post-processing and test different methods for the generation of binary saturation maps from the TIR images. We test the methods on various images with different image characteristics. Results show that the best method, in addition to a manual image classification, is a statistical approach that combines the fitting of two pixel class distributions, adaptive thresholding, and region growing.

## 1 Introduction

The patterns and dynamics of surface saturation areas have been on hydrological research agendas ever since the formulation of the variable source area (VSA) concept by Hewlett and Hibbert (1967). Surface saturation is relevant for runoff generation and for water quality, due to variable active and contributing areas (Ambroise, 2004), as well as critical source areas (e.g. Doppler et al., 2014; Frey et al., 2009; Heathwaite et al., 2005). Likewise, surface saturation patterns and their dynamics are

closely linked to groundwater-surface water interactions (e.g. Frei et al., 2010; Latron and Gallart, 2007) and catchment storage characteristics and dynamics (e.g. Soulsby et al., 2016; Whiting and Godsey, 2016).

Despite the prominent role of saturated areas in hydrological processes research, mapping them remains a challenging exercise. The most straightforward mapping method consists of locating saturated areas by walking through the catchment. However,

this simple but labour-intensive 'squishy-boot' method (e.g. Blazkova et al., 2002; Creed et al., 2003; Latron and Gallart, 2007; Rinderer et al., 2012) is suitable neither for large areas nor for fine-scale spatial resolutions. Dunne et al. (1975) introduced topography, soil morphology, hydrometric measurements (soil moisture, water table level, baseflow), and vegetation as useful indicators for delineating saturated areas. It is still a valid research question today of how to best make use of these catchment characteristics to delineate saturated areas (e.g. Ali et al., 2014; Doppler et al., 2014; Grabs et al., 2009; Kulasova et al., 2014a,

2014b). Hydrometric measurements offer the potential for monitoring the local temporal evolution (in increments ranging from minutes to months) of dynamic surface saturation. The analysis of topography, soil morphology, or vegetation allows lasting saturation patterns to be identified for large contiguous areas.

Remote sensing has proven to be well-suited for mapping temporal dynamic patterns of surface saturation over large areas. It is possible to extract flooded areas in the order of metres to kilometres from data acquired with satellite and airborne platforms,

such as synthetic aperture radar (SAR) images (e.g. Matgen et al., 2006; Verhoest et al., 1998), or the normalized difference water index (NDWI) and the normalized difference vegetation index (NDVI) (de Alwis et al., 2007; Mengistu and Spence, 2016). Observations at higher spatial resolution (order of centimetres) require unmanned aerial vehicles (UAVs) or ground-based instruments. Due to various technical constraints, to date, SAR image acquisitions are rarely used for UAV-based applications or for ground-based applications that are not restricted to a fixed location (e.g. Li and Ling, 2015; Luzi, 2010).

NDWI and NDVI are applicable at these scales (e.g. Orillo et al., 2017; Wahab et al., 2018), however, to the best of our knowledge, the necessary simultaneous acquisition of short-wave infrared and visible light (VIS) images has not yet been performed by UAVs or on the ground for mapping surface saturation.

Ishaq and Huff (1974) and Dunne et al. (1975) suggested the use of VIS or infrared photographs for mapping surface saturation. However, this suggestion has rarely been followed in the last 40 years (with Portmann, 1997, being a notable exception) despite

VIS cameras having been deployed on the ground and mounted on UAVs, airborne or satellite platforms for a long time. Recently, Chabot and Bird (2013) and Spence and Mengistu (2016) successfully used VIS cameras mounted on UAVs for mapping surface water (a wetland of 128 ha and an intermittent stream surveyed via three transects of 2 km each). Silasari et al. (2017) mapped surface-saturated areas on an agricultural field (100 m x 15 m) using a VIS camera mounted on a weather station for high-frequency image acquisition.

Since the advent of affordable, handheld thermal infrared (TIR) cameras, TIR imagery features the same temporal and spatial flexibility as VIS imagery. In the context of this technical advancement, TIR imagery started to be used for analysing hydrological processes such as groundwater – surface (water) interactions (e.g. Ala-aho et al., 2015; Briggs et al., 2016; Pfister et al., 2010; Schuetz and Weiler, 2011) or water flow paths, velocities, and mixing (e.g. Antonelli et al., 2017; Deitchman and Loheide, 2009; Schuetz et al., 2012). However, applications of TIR imagery for mapping surface saturation are rare. Two

examples are from Pfister et al. (2010) and Glaser et al. (2016), who demonstrated the potential for TIR imagery to map surface saturation by carrying out repeated TIR image acquisitions at small spatial scales (centimetres to metres) with handheld cameras.

One reason for the scarce number of studies that use TIR imagery for mapping surface saturation is certainly that few descriptions of the methodological advantages and challenges exist. However, there are several general guidelines and methodological descriptions for TIR imagery applications. These studies focus on one specific aspect of TIR imagery, such as co-registration (Turner et al., 2014; Weber et al., 2015) or on how to acquire correct surface water temperatures, which is the most common application of TIR imagery in hydrology (e.g. Dugdale, 2016; Handcock et al., 2006, 2012; Torgersen et al., 2001). Many of these recommendations can be directly applied for mapping surface saturation via TIR imagery (e.g. choice of sensor type). However, some recommendations are redundant (e.g. temperature corrections) or different (e.g. optimal time scheduling) for the application of TIR imagery for surface saturation mapping.

Here, we go beyond the mere demonstration of the potential for TIR imagery to map saturated surface areas and address the related application-specific technical and methodological challenges. The novelty of this work is that we assimilate, within one study, fundamental principles, technical aspects, and methodological possibilities and challenges with an exclusive focus on the mapping of surface saturation. This includes all steps, from image acquisition to the generation of binary saturation maps. To do this, we (1) review relevant technical and methodological aspects from existing TIR imagery literature and (2) complement them with our expertise and results from an 18-month field campaign. The field campaign focused on the recurrent acquisition of panoramic images with a portable TIR camera in seven distinct riparian areas. The precautions and considerations that we describe in this technical note are also valid for surface saturation mapping campaigns with permanently installed ground-based TIR cameras and TIR cameras mounted on UAVs, and airborne or satellite platforms.

The manuscript is structured in two main parts. The first part (Section 2) focusses on the mapping approach itself and combines a literature review with examples of our own experience. The second part (Section 3) demonstrates the application of different pixel classification techniques for generating binary saturation maps from TIR images by applying and comparing them for different example images. A discussion and a conclusion section evaluate the key features of the manuscript and outline perspectives for future research and applications for TIR imagery in hydrological sciences.

## 2 Mapping surface saturation with TIR imagery: state of the art and examples

### 2.1 Fundamental principles

TIR cameras are used for measuring surface temperatures remotely (e.g. 100 µm penetration depth for water columns) within an area of interest. The cameras sense the intensity of thermal infrared radiation emitted by the objects the camera is pointed at. The surface temperature $T$ (K) of the objects is then calculated from the sensed radiant intensity $W$ (Wm$^{-2}$), based on Stefan-Boltzmann's law with the Stefan-Boltzmann constant $\sigma = 5.67*10^{-8}$ Wm$^{-2}$K$^{-4}$

$$T = \sqrt[4]{(W/\sigma)} \qquad (1)$$

Considering radiometric corrections for material-specific emissivity ε, for reflections of radiation from the surroundings, and for atmospheric induced and attenuated radiation, the radiant intensity $W$ is split into the emission from the object ($W_{obj}$), from the ambient sources ($W_{refl}$), and the atmosphere ($W_{atm}$)

$$W = \varepsilon\tau W_{obj} + (1 - \varepsilon)\tau W_{refl} + (1 - \tau)W_{atm} \qquad (2)$$

with τ being the transmittance of the atmosphere, which depends on the distance between the object and the camera sensor, as well as on relative air humidity. Ultimately, values for the temperature of the ambient sources and the atmosphere, the targeted object's emissivity, the distance between object and camera, and the relative humidity are required for accurately estimating an object's surface temperature $T$.

Details on the principles of TIR imagery, TIR sensor types (i.e. wave length, sensitivity), and considerations for choosing the most appropriate camera and remote sensing platform for the desired acquisition (i.e. accuracy, resolution) are provided in the literature (cf. Dugdale, 2016; Handcock et al., 2012). For this study, we relied on two different handheld TIR camera models: a FLIR B425 with a resolution of 320 x 240 pixels and an angle of view of 25° and a FLIR T640 with a resolution of 640 x 480 pixels and an angle of view of 45° (FLIR Systems, Wilsonville. USA). The wider angle of view of the FLIR T640 clearly

facilitated the image acquisition in this study, while a pixel resolution lower than the resolutions of the two cameras would still have been sufficient for the identification of surface saturation patterns.

We define surface saturation as water ponding or flowing on the ground surface (even if only present as a very thin layer). Mapping surface saturation with TIR imagery requires (1) a sufficient temperature contrast between surface water and the surrounding environment (e.g. dry soil, rock, vegetation) and (2) at least one pixel of the TIR image being known to correspond

to surface water. When these two requirements are met, it is possible to visually identify the surface saturation patterns in a TIR image. This is exemplified with a TIR image of a riparian-stream zone (Fig. 1). The substantial temperature contrast (requirement 1) allows us to differentiate between two TIR pixel groups, i.e. surface water pixels and surrounding environment pixels. With ground truth data at hand (here: VIS image, alternatives include stream water temperature or knowing the location of the creek) for point 1 of Fig. 1 (requirement 2), the group of pixels with higher temperatures can be identified as surface

water. The group of pixels with lower temperatures can be regarded as non-saturated surrounding environment (cf. Fig. 1, point 2). With this classification in mind, the TIR image significantly amplifies the appearance of surface-saturated areas relative to a VIS image. Moreover, the TIR image reveals additional surface-saturated areas that are not clearly identifiable (cf. points 3 in Fig.1) or not visible (cf. area above point 6, Fig.1) within a VIS image.

The example shows that the identification of surface saturation relies on temperature contrasts between surface water and the

surrounding environment. Radiometric corrections of TIR images for obtaining correct temperature values are thus not necessary. However, interferences that affect temperature, such as shadow casts or reflections (cf. Dugdale, 2016; Handcock et al., 2012), cannot be disregarded as they can influence the temperature contrast (see Section 2.2). In cases where the water temperature is too similar to the surrounding materials, saturated areas might be falsely identified as dry, whereas surrounding

materials might be falsely identified as wet. In cases where non-uniform water temperatures occur, different water sources may be distinguished (cf. Fig.1, where point 4 likely represents stream water, points 5 and 7 likely represent the exfiltration of warmer groundwater). However, a bimodal distribution of water temperatures (e.g. cold stream and warm exfiltrating groundwater or warm ponding water) can also lead to a misinterpretation of temperature contrasts to the surrounding environment (e.g. a surrounding material with a temperature that is in-between the water temperatures might be identified as water).

For the above-mentioned reasons, it is important to evaluate the applicability of the TIR images for identifying the surface-saturated areas with some ground truth / validation data. For the validation, we relied on immediate visual verification during image acquisition, as well as on VIS images. Another option is to install sensors that can verify the presence or absence of water on the ground surface locally, yet this is an experimental effort and only results in validation data for selective points. Validating the TIR images with other saturation mapping techniques is difficult, since most of these techniques implicitly include saturation in the upper soil layer, while the current use of TIR imagery excludes the soil. For example, saturated areas inferred via the squishy boot method account for areas where water is squeezed out of the soil when stepping on it, whereas such areas are not detected as saturated areas by the non-invasive TIR imagery.

## 2.2 Image acquisition interferences

*Impact of weather conditions*

Weather conditions can interfere with TIR image acquisition (e.g. Dugdale, 2016; Handcock et al., 2012). The main problem stems from similar temperatures of water and the surrounding environment, compromising an identification of surface saturation with TIR images (Fig. 2a). Water has a higher thermal capacity than most environmental materials and therefore the water surface temperature generally aligns more slowly with the air temperature than the surface temperatures of surrounding materials. During our field campaign, it became clear that particularly during day-night-day or seasonal transitions, this difference in thermal capacities induced a convergence of the surrounding environment's temperatures (which align to the air temperature) to the water temperature. Furthermore, direct exposure of the study site to sunlight, combined with shadow casts, commonly distorted the temperature contrasts. Surrounding materials in the shade with temperatures different to the same surrounding materials in sunlight led to reduced temperature contrasts between these materials and the surface water (Fig. 2b). Once the direct sun exposure ceased, the different thermal capacities of different materials heated by the sun could still cause patches of warmer and colder temperatures. Rain and fog may also influence image quality due to water droplets falling between the TIR sensor and the ground, eventually blurring the images and causing uniform temperature signatures (Fig. 2c). To avoid the acquisition of non-useable TIR images, we advise planning field campaigns adapted to the weather forecasts. The ideal situation is to work during dry weather with warm or cold air temperatures in order to ensure a clear difference between the temperature of the surrounding materials and the more temperate water surface temperatures. Dugdale (2016) reported the time period from mid-afternoon to night-time as an optimal TIR image acquisition period for monitoring water surface temperatures. Based on our 18-month field campaign, we suggest that the optimal TIR image acquisition time for identifying

surface saturation patterns is early morning. At this time, there are no undesirable effects due to sunlight (shadows, warming-up) and there are generally high temperature contrasts between water surfaces and the surrounding environment. Cloudy conditions can also help to avoid the effect of direct sunlight. A site-specific analysis of the sun exposure throughout the day can help pinpoint at what other times images can be taken in favourable conditions for a specific study site.

*Camera position*

Obstructions in the TIR camera's field of view are obviously problematic. Yet, permanent view obstructions on the ground (e.g. tree trunks, Fig. 2d, point 6) proved to be useful ground reference points during our field campaign. Temporary view obstructions, such as growing vegetation (Fig. 2d), recent litter, and snow cover are a problem for repeated imaging campaigns.

Cutting the vegetation during the growing season is an option for small study sites. Our experience is that the coverage of grasses and herbaceous plants with small leaves is normally low enough to permit the recording of the ground surface temperature, while the coverage of ferns or tree leaves is normally completely opaque. Snow cover usually hides surface saturation. Yet, periods where the amount of snow is low are commonly not problematic, since the saturated areas mainly stay uncovered due to a warmer water temperature and thus fast melting of the snow.

Ideally, images are taken from above and at nadir to the study site. Oblique angles of view (>30° of nadir) reduce the object's emissivity and thus distort the detected temperatures in the TIR images (Dugdale, 2016). The incorrect temperature values are not critical as such for mapping surface saturation patterns, but we observed that wide ranges of angles can result in distinct temperature distortions and thus reduced temperature contrasts within the images. In a similar way, varying distances between camera and ground surface for different positions within one image (e.g. top / bottom, left / right) do not only provoke pixels

with varying area equivalents, but can also distort the temperature detection and thus temperature contrast. Therefore, ground-based cameras should be positioned at locations that minimize the range of angles of view and the distances between camera and ground surface. In the event of repeated image acquisitions of a given area of interest, we took the pictures from the same position each time in order to facilitate subsequent image comparisons. For repeated image campaigns, it could be useful to install a structure that allows several images to be acquired by moving the camera to specific positions with fixed heights

above the ground and fixed angles of view. This could simplify the post-processing and assemblage of the images into panoramic images (cf. Section 2.3).

*Measurement artefacts during image acquisition*

For determining surface saturation, the TIR images should cover an area known to be surface-saturated (e.g. stream, visually

obvious wet spots) in order to have a reference for water temperature (cf. Section 2.1). In addition, a VIS image should be acquired simultaneously to the TIR image for comparison. The TIR imagery parameters necessary for correcting and converting the radiation signal to temperature values (e.g. air temperature, humidity) do not need to correspond to the actual conditions, since only the temperature contrast and not the correct temperature value is required for defining saturated areas. Certainly, 'wrong' temperatures influence the temperature contrast between the surroundings and the water, but this effect on

the contrast can be negative or positive. If correct temperatures are targeted, radiometric corrections need to be applied during the image post-processing procedure. This allows for example to consider different emissivities for different surface materials by using appropriate values for each individual image pixel (Aubry-Wake et al., 2015). However, in our experience, setting realistic parameter values during the image acquisition helped the auto-focus process of the camera and prevented the

observation of unrealistic surface temperatures. Nonetheless, in the event of clear skies or on cold winter days, we occasionally observed negative temperatures for flowing water. The explanations for these observations remain speculative. Potentially, a particularly strong reflection of the radiation from the surroundings and the sky in the water influenced the temperature detection. However, for the identification of surface saturation patterns, such unrealistic negative temperatures do not pose a problem since the temperature range stays correct (Antonelli et al., 2017).

Reflections of surrounding objects on the water surface (Fig. 2e) and image vignetting can occur during image acquisition and can compromise a further use of the TIR images. Vignetting is the falloff of radiation intensity towards the edges of the image, which is mainly generated by the geometry of the sensor optics (esp. wide angle lenses) (cf. Kelcey and Lucieer, 2012). As a consequence, the monitored temperature can change towards the edges of the picture (cf. aura effect in Antonelli et al., 2017). In this study, the image vignetting was unproblematic, especially where a panorama was built from several images (cf. Section

2.3). This is due to the fact that the effect of image vignetting only occurs at the edges of the pictures and it is of minor relevance in images with high temperature contrasts. Reflections of surrounding objects on the water surface limit the value of the images for saturation identifications in a similar way to shadows (cf. Fig. 2d and 2e). The difference with shadows is that reflections also occur with diffuse light, which makes it difficult to predict their occurrence and thus to avoid them.

**2.3 Generation of TIR panorama images**

We acquired the images used for the assemblage of a panoramic view in two different ways: (1) by taking single, overlapping images, and by (2) taking a video of the area of interest. While both approaches deliver similar final results, videos are recorded faster than sequences of individual images. Independently from the chosen data format, we ensured that the saving format retained the temperature information as radiometric data for further image processing (see below and Fig. 3). Sun (dis)appearance and automatic noise corrections by the camera (non-uniformity corrections, cf. Dugdale, 2016) can lead to

considerable shifts in recorded temperatures from one image / video frame to another. Since correcting such temperature shifts is difficult (cf. Dugdale, 2016), we opted to control them by fixing the temperature – colour scale and restarting image acquisition if the colour (and thus temperature) of overlapping image parts changed.

We acquired the images / video frames in such a way that the area of interest formed the central part of a panorama. This allowed us to avoid image gaps and distortion effects at the borders of the area of interest. When possible, we ensured that the

single pictures / video frames included overlapping parts with identifiable structures such as the stream bank, tree stems, or stones as natural reference points. For videos, it was essential to move the camera slowly enough to obtain sharp images and to use a low frame rate (e.g. 2 Hz) to keep the number of video frames reasonable (enough frames for obtaining area overlaps, but not too many frames showing the same area).

The generation of a panorama from overlapping TIR images / video frames acquired with a ground-based camera involves some challenges that specifically relate to TIR and / or ground-based images. This needs to be addressed in TIR-specific panorama generation and image processing steps, as presented briefly by Cardenas et al. (2014). Our approach consisted of transforming the acquired images / video frames containing the radiometric information (see above) into grey-scaled, standard

format images / videos (Fig.3, step 1) in order to allow the use of ordinary panorama assemblage software. We relied on grey colour scale images, linearly splitting the colour shades over the global temperature range of the acquired images / video frames, since this prevents artefactual colour mixing effects and allowed us to embed the temperature information in the generated panoramas. When the extreme temperature values of an image were not relevant for the identification of saturated areas, we truncated the global temperature range in favour of a better colour contrast and a finer temperature class width

retained in the grey values (e.g. the retained temperature class width is 0.1 °C in case of a temperature range of 25.5 °C and an image with 255 grey values).

We employed Microsoft's Image Composite Editor (ICE) and the PTGui panorama software (New House Internet Services) to create panorama images (Fig.3, step 2). ICE and PTGui allow the creation of panoramas from single images (and from video frames for ICE) with an automatic mosaicking function (i.e. a function that geometrically transforms, aligns, and overlaps the

single images). TIR images generally show less identifiable features and lower contrasts than VIS images (cf. Weber et al., 2015). Therefore, a (partial) failure of automatic mosaicking is not uncommon and manual interactions with image alignment (i.e. defining control points for matching distinct points in overlapping images in PTGui) were frequently necessary for the TIR images taken during our 18-month field campaign.

In order to compare several panorama images of the same area, one needs to co-register the panoramas (Fig. 3, step 3). In

principle, it is possible to geo-rectify the TIR images by allocating geographical coordinates to the images, which are derived from ground control points (cf. Keys et al., 2016; Silasari et al., 2017) or from a virtually projected elevation model (cf. Cardenas et al., 2014; Corripio, 2004; Härer et al., 2013). However, this can result in large gaps or strong interpolations and distortions in the images, due to view obstructions in the picture. Instead of this, therefore, we co-registered TIR panoramas of the same area against each other (cf. Cardenas et al., 2014; Glaser et al., 2016). More specifically, we registered and cropped

them to the dimensions of a reference TIR panorama of the area of interest (Fig. 3, step 3).

## 2.4 Application examples

In this section, we present three examples from our 18-month field campaign that demonstrate the potential for TIR imagery to analyse surface saturation patterns and their dynamics. All images were taken in the Weierbach catchment - a forested, 42 ha headwater research catchment in western Luxembourg (Glaser et al., 2016; Klaus et al., 2015; Martínez-Carreras et al.,

2016; Schwab et al., 2018). We avoided unfavourable environmental conditions for the image acquisitions (cf. 2.2, Fig. 2) by allowing a few days tolerance around the targeted (bi-)weekly recurrence frequency. Additionally, we cut ferns that obstructed the camera view during the summer months. The 364 acquired panorama images were divided into three groups classified as

usable without restrictions (32.4 %), usable with some restrictions (small negative effects of low temperature contrasts or covering vegetation visible, 31.1 %), and unusable (36.5 %).

The usable panoramas captured the temporal evolution of surface saturation over the 18-month field campaign. This demonstrates the robustness of TIR imagery through the complete range of seasonal conditions (Fig. 4), including snow and growing vegetation, as well as warm and cold water. The full extent of added value provided by TIR imagery compared to VIS imagery was documented for cases with different seasonal conditions (Fig 4 a/e vs. b vs. c/d), particularly for situations with less pronounced differences in discharge levels (Fig. 4 a vs. b vs. c, d vs. e). For example, the comparison of the VIS images of December 2015 and June 2016 (Fig. 4 a vs. c) suggests wetter conditions for December 2015, while the two TIR images show similar saturation patterns for the two dates.

In addition to surface saturation dynamics, the TIR images can also reveal distinct types of saturation patterns. For example, the orientation of saturated areas may change over a few metres from perpendicular (Fig. 5 top) to parallel (Fig. 5 bottom) to the adjacent stream. The extension of saturated areas along the left bank (Fig. 5 bottom) appears to be created by a parallel extension of the stream in a flat riparian zone that becomes an extended stream bed. The surface saturation oriented perpendicularly to the stream at the right bank (Fig. 5) appears to be generated from exfiltrating groundwater that flows downhill to the stream at the soil surface. Thus, the different directional extents of the saturated areas can indicate different processes underlying the surface saturation formation.

Finally, the images allow us to identify the spatial heterogeneity of temporal saturation dynamics across different study sites. Figure 6 shows TIR images of the riparian zone of two different source areas with different degrees and dynamics of surface saturation. In area 1 (Fig. 6, left panels), the pattern of saturation areas barely changed from February to April, while in area 2 (Fig. 6, right panels) some locations had dried out (red circles). In December 2016, the riparian zones of both source areas were completely dry and the stream started further downstream in comparison to the other observation dates (red arrows). This suggests that both source areas evolve from very wet to very dry conditions (during which surface saturation is mainly represented by spots with stable groundwater exfiltration) with distinctly different transition dynamics.

## 3 Quantification of saturation through pixel classification

### 3.1 Methods for generating binary saturation maps

The application examples described in Section 2.4 demonstrate the potential for TIR images to rapidly and intuitively visualize surface-saturated areas. However, the 'raw data' images need to be transformed into binary saturation maps for further analyses based on quantitative values (as e.g. saturation percentages). A common approach to binarizing an image is histogram thresholding (e.g. Rosin, 2002). This allows a TIR image to be transformed into a binary saturation map by taking the temperature range of pixels that are known to be saturated (i.e. stream pixels) and defining all pixels in that image that fall into that temperature range as saturated (cf. Glaser et al., 2016; Pfister et al., 2010). Several thresholding algorithms can be found in the literature, each of which has its characteristic assumptions with respect to image content (Patra et al., 2011).

Unsupervised approaches other than thresholding are also used for image binarization, e.g. clustering (Li et al., 2015). Yet, thresholding is the most rapid technique for achieving a binary classification of an image, even though the selection of an adequate threshold value represents a critical step and its choice strongly influences the classification outcome.

One possibility to select a threshold value for classifying surface saturation is to manually adapt the temperature range until the resulting saturation map matches best the visual assessment of the original TIR and – if possible – VIS image. A more objective and, for time-lapsed images, faster method consists of relying on the temperature of preselected pixels or a predefined mask for saturated and unsaturated parts in all images. Such pixels / masks can be selected based on a visual interpretation of the images or on information obtained from reference sensors in the field, indicating whether a location was wet or dry at the surface at the time of image acquisition.

Silasari et al. (2017) applied an automatic image classification for unimodal distributions based on a threshold parameter that needs to be calibrated to specific image conditions (in this case, the brightness of VIS images).This is only straightforward in cases where the temperature distribution between water and the surrounding environment is clearly bimodal. Chini et al. (2017) presented a parametric adaptive thresholding algorithm especially suited for images that do not show a clear bimodal distribution. The algorithm makes use of an automatic selection of image subsections with clear bimodal distributions, a hierarchical split-based approach (HSBA), and a subsequent parameterization of the distributions of the two pixel classes. Since the two decomposed distributions might still overlap to a certain extent, Chini et al. (2017) advise complementing the decomposed distribution information with contextual information of the image for the final generation of a binary image instead of selecting a single threshold value between the two decomposed distributions. Several approaches are available in the literature for including contextual information in the classification of a single spectral image, such as mathematical morphology (Chini et al., 2009) or second order textural parameters (Pacifici et al., 2009). Chini et al. (2017) suggested a region growing algorithm where the seeds and the stopping criteria are constrained by the identified distribution of the class of interest (i.e. here saturation).

## 3.2 Comparison of methods for generating binary saturation maps for TIR images

We applied three of the approaches described above to generate binary saturation maps on our TIR image data set. Here, we present the results for four example images with differing conditions during image acquisition (e.g. very wet/dry conditions, water being the warmest / coldest material, Fig. 7). We evaluated the results of the three different approaches based on our observations from the field and the corresponding VIS image as ground truth.

First, we manually chose a temperature range of saturation for each image. By its nature, this pixel classification approach creates results that are very close to ground truth. However, finding an unequivocal temperature range was not feasible and the selection of the most plausible temperature range (Fig. 7, dark-green asterisk) remained somewhat subjective. Furthermore, artefacts (such as pixels corresponding to vegetation covering the stream) induced some uncertainty in the pixel classification, eventually leading to discrepancies compared to visually identified saturation patterns. Consequently, a pixel classification based on this manual procedure remained tarnished by some uncertainties. The definition of an uncertainty range within which

the temperature range can be considered plausible (Fig. 7, dark-green, dashed lines) was also subjective. Generally, the uncertainty range was small for images with low saturation and gradually increased with higher saturation (compare Fig. 7 d-b). Accordingly, images with a large difference in percentages of saturated pixels (e.g. Fig. 7b vs. 7d) did not encounter an overlap of the uncertainty ranges. For some images, the uncertainty range was rather high (Fig. 7a) and a comparison with other images with percentages of saturated pixels in the same range was thus problematic. In such cases, it is preferable that only one person defines the optimal temperature ranges and thus saturation patterns for all images that are intended to be compared in order to ensure consistency in the image interpretation.

Secondly, we performed an objective selection of the temperature range of saturation based on masks with known pixel classes. For this, we used two masks, one with 2000 pixels falling into an area that always stayed dry and one with 2000 pixels falling into an area where the stream was flowing all year (red rectangles, Fig.7). Based on the mask, we selected the threshold for the temperature range as the 90$^{th}$ percentile and 10$^{th}$ percentile of the temperature of the stream mask pixels and dry mask pixels, respectively (i.e. 90% of the pixels falling below the mask were defined as saturated and dry, respectively). By using the two different masks, we obtained two temperature ranges, resulting in two different saturation percentages for each image (Fig. 7, blue points). The identification of saturated areas based on the dry mask was clearly not constrained enough. The identification of saturated areas based on the stream mask sometimes approached the manual identification of saturation (Fig. 7 a, c) but in other cases even exceeded it (Fig. 7d). The uncertainty range of saturation obtained with the two masks could be reduced by selecting a more extreme percentile for the temperature threshold definition. However, this increased the risk of obtaining a clearly incorrect value (cf. Fig. 7d), since the stream / dry mask can cover pixels of the 'wrong' category (due to artefacts like vegetation covering the stream or due to distorted co-registered images, resulting in a shifted mask). A reduced mask size prevents such 'wrong' pixels, but also reduces the captured variability in temperature (in an extreme case down to one temperature value), which in turn increases the risk of missing the warmest or coldest temperature of the water or dry areas.

Finally, we tested the usability of the approach proposed by Chini et al. (2017), constraining a region growing algorithm to a) a bimodal distribution derived from the HSBA applied to the entire image, b) a bimodal distribution derived from the HSBA where the selection of bimodal image subsections was constrained to image-specific manual predefinitions of temperature ranges of saturation, and c) a bimodal distribution derived from pre-selected parts of the image that include clearly wet and dry areas. While in some cases the fully automatic image classification (a) worked very well in comparison to the manual selection of a temperature range (cf. Fig. 8 04/12/15, 30/08/16), for the other cases, saturation was mostly underestimated (cf. Fig. 8 25/02/16, 03/06/16). The additional constraint with image-specific temperature ranges (b) overall improved the matches with the manually defined saturation patterns, but the result was strongly influenced by the match of the given constraint range to the range that was defined as the optimum for the image. A constraint with a rough estimated temperature for saturation worked less well than a constraint with the temperature range as selected in the detailed manual assessment described earlier in the Section (cf. Fig. 7 green asterisks and lines). The classification based on pre-selected parts of the image (c) tended to result in higher saturation amounts. This improved the match for the cases that were underestimated with the fully automatic

classification (a) (cf. Fig. 8 25/02/15, 03/06/16), but overestimated saturation for the cases where the fully automatic classification (a) showed good results (cf. Fig. 8 04/12/15, 30/08/16).

## 4 Discussion

### 4.1 Mapping surface saturation with TIR imagery

The main advantages of TIR imagery in comparison to other surface saturation mapping methods are its non-invasive character and its large temporal and spatial flexibility (centimetres to kilometres, minutes to months). Another advantage is that TIR images allow a rapid and intuitive identification and analysis of the dynamics of surface saturation patterns. The 'raw data' images can be used without any additional processing to study surface-saturated areas, their evolution over time, and how and where they occur – ultimately contributing to a better mechanistic understanding of the hydrological processes prevailing in

the studied area. The pure visual information provided *per se* by the images is also usable as soft data, e.g. for model validation (e.g. different types of extent compared to stream, Fig.5, more and less stable saturation patterns, Fig.6). VIS imagery offers similar advantages (Silasari et al., 2017), but commonly the saturated areas are not as clearly visible as with TIR imagery (cf. Fig. 1, Fig. 4). Moreover, VIS imagery is not usable during the night and cannot provide additional information about water sources and processes underlying the surface saturation formation (cf. Fig. 1, Fig. 5, groundwater inflow vs. stream water).

Nevertheless, VIS imagery provides good complementary information to the TIR imagery and should always be considered as a ground truth information source.

In our study, unfavourable image acquisition conditions (cf. Section 2.2) caused 36.5 % of the acquired images to be unusable for further processing. High amounts of unusable images are a common problem in environmental imagery (cf. e.g. cloud cover for satellite images, night-time for VIS images (DeAlwis et al., 2007; Silasari et al., 2017)). Flexibility in the scheduling

of a field campaign is thus necessary to reduce the number of acquisitions during unfavourable conditions. A concern for the use of TIR imagery for mapping saturation patterns is that some saturated areas (e.g. warmed-up ponding water) might not be identified as saturated due to a temperature that is very different from stream temperature. This relates back to the fact that temperature is only used as an indicator for saturation. Compared to other saturation indicators, such as vegetation mapping or hydrometric measurements (cf. Dunne et al., 1975), we consider TIR imagery with the above-mentioned advantages as the

better indirect mapping method. However, the only way to directly map surface saturation consists of walking through the area of interest (e.g. squishy boot method), which remains restricted to small areas and / or low mapping frequencies.

The amount of fieldwork for imagery mapping is generally reduced compared to other methods for mapping surface saturation (e.g. vegetation/soil mapping), allowing more frequent campaigns with higher spatial precision. Yet, in consistency with other imagery mapping studies (e.g. Spence and Mengistu, 2016), the image post-processing in this study was time-consuming.

Mosaicking and the co-registering of images is often considered particularly difficult for TIR images, since ground control points with a thermal signature are needed (Dugdale, 2016; Weber et al., 2015). Our experience showed that the images normally offered enough natural thermal ground control points (e.g. the stream bank) in cases where the temperature contrast

between water and ambient materials was good enough for image usability. In combination with the post-processing workflow presented, the post-processing effort was reasonable. More automatized workflows like the one proposed by Turner et al. (2014) for mosaicking UAV acquired TIR images could also be adapted and applied.

The image acquisition considerations, post-processing steps, and application examples described focused on (bi-) weekly
panoramic images of small areas, acquired with a portable TIR camera. A transfer of the TIR imagery technique to different temporal or spatial scales does not change the principles and possibilities of the technique, but will require some additional scale- and platform-dependent considerations. For example, using permanently installed ground-based cameras for image acquisitions with high temporal frequencies might challenge technical aspects such as protection of the camera against environmental influences, an automatic triggering of image acquisition, and power supply. These aspects might also be relevant
for TIR imagery acquisition at larger spatial scales, especially when using UAVs. Besides, image acquisitions based on UAV or aeroplane overflights might for example require considerations of overflight regulations, saturation patterns within a forest could only – if at all – be mapped during the dormant season, and ground control points and ground truth data might be more difficult to obtain. Partly, such challenges are addressed in existing literature (e.g. Vivoni et al., 2014; Weber et al., 2015), others will need to be figured out by applying the TIR technique at such different scales.

### 4.2 Pixel classification methods

More challenging than TIR image mosaicking and co-registering was the generation of saturation maps from the TIR images. The different pixel classification methods tested all yielded somewhat different results compared to pixel classification based on manual visual assessment. Nevertheless, realizing an objective, automatic classification of saturated areas is not more challenging than for other surface saturation mapping methods. Saturation maps created based on the squishy boot method or
vegetation/soil mapping are subjective due to decisions made during the fieldwork. The (un)supervised classification methods that are commonly used for creating saturation maps from remote sensing data (e.g. VIS images / NDVI/NDWI) also contain some uncertainty (Chabot and Bird, 2013; DeAlwis et al., 2007; Mengistu and Spence, 2016; Spence and Mengistu, 2016). Moreover, the main problem for all of the tested saturation map generation methods (cf. Section 4) is that they are not applicable without adapting them to individual image conditions (very wet, very dry, water being the warmest / coldest
material, slightly different field of views). Other image processing methods for deriving saturation maps also do not fulfil this requirement and it is necessary to adapt the parameters (e.g. Silasari et al., 2017) or do a new supervision (with new classification pixels/masks) for the classification of images with different conditions (e.g. Chabot and Bird, 2013; Keys et al., 2016). At this stage, we consider a manual choice of temperature range for saturated pixels the best approach for time-lapsed images with very variable conditions and slight perspective shifts, even though it is labour-intensive and somewhat subjective.
For time-lapsed images with a fixed vantage point and for time spans with similar conditions (e.g. storm events), the automatable methods presented represent valuable options. In particular, the combination of an automatic decomposition of two pixel class distributions with a region growing algorithm yielded objective saturation maps close to the manual saturation classification and visual assessment of the TIR images (Fig. 8). Small adaptations of the constraint for the decomposition of

two pixel class distributions were sufficient to obtain good results for the different image conditions (cf. Fig. 8 a – c) and further developments of the method might even allow such adaptations to be performed in a (semi)automatic way. More work on pixel classification might also include the application of machine learning techniques or, especially for time-lapsed images, the analysis of the temperature signals of individual pixels over time. Another interesting option may consist of combining the

TIR images with additional data (e.g. VIS images or NIR images), which will allow multi-spectral classification methods to be applied (Chini et al., 2008) and at the same time, contextual information to be integrated (Chini et al., 2014).

## 5 Summary and conclusions

This technical note presents recent work carried out in the Weierbach catchment, where we tested the potential for TIR imagery to map surface saturation dynamics. For the best of our knowledge, this is the first comprehensive review and summary of the

TIR imagery-related methodological principles and the required precautions and considerations for a successful application of TIR imagery for mapping surface saturation. We give advice for all steps, from image acquisition to processed saturation maps. The main requirement is a clear temperature contrast between water and the surrounding environments. Image acquisition during an 18-month campaign showed that the method works best during dry nights or dry early mornings and that images should be taken from well-chosen positions without obstructions in view towards the ground. The workflow presented for

acquiring panoramic images is particularly suitable for small areas of interest (centimetres to metres) that are monitored with intermediate to low mapping frequencies (days to months). Moreover, the information contained in this technical note is also beneficial for applications at different temporal and spatial scales (fixed cameras for high frequency images, drone/satellite images for larger spatial scales), considering that some adaptions and further developments of the methodology might be necessary.

We demonstrated with three examples that TIR imagery is applicable throughout the year and can reveal spatially heterogeneous surface saturation dynamics and distinct types of saturation patterns. The saturation patterns can also be used to identify different processes underlying the surface saturation formation, such as groundwater exfiltration or stream expansion. The surface saturation information visualized in the images can be used directly as soft data for characterizing field conditions, for analysing ongoing hydrologic processes and for model validation.

The methods presented for obtaining binary, objective saturation maps from TIR images contain some uncertainties and are not automatable for datasets containing many images with varying characteristics (e.g. very wet / dry, water warmest / coldest material, slightly different field of views). In such cases, a manual choice of the temperature range for saturated pixels is the most reliable approach. Yet, for image subsets with similar conditions, the pixel classifications tested work well and we think that the combination of an automatic decomposition of the image distribution in two pixel classes and a region growing

algorithm is a very promising option for obtaining objective, comparable saturation maps. In conclusion, we consider the TIR imagery a very powerful method for mapping surface saturation in terms of practicability and spatial and temporal flexibility

and we believe it can provide new insights into the role of saturated areas and subsequent spatial and temporal dynamics in rainfall runoff transformation.

**Acknowledgments**

We would like to thank the three anonymous reviewers whose suggestions helped to improve the manuscript. Barbara Glaser thanks the Luxembourg National Research Fund (FNR) for funding within the framework of the FNR-AFR Pathfinder project (ID 10189601). Marta Antonelli was funded by the European Union's Seventh Framework Programme for research, technological development and demonstration under grant agreement no. 607150 (FP7-PEOPLE-2013-ITN – INTERFACES – Ecohydrological interfaces as critical hotspots for transformations of ecosystem exchange fluxes and bio-geochemical cycling).

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

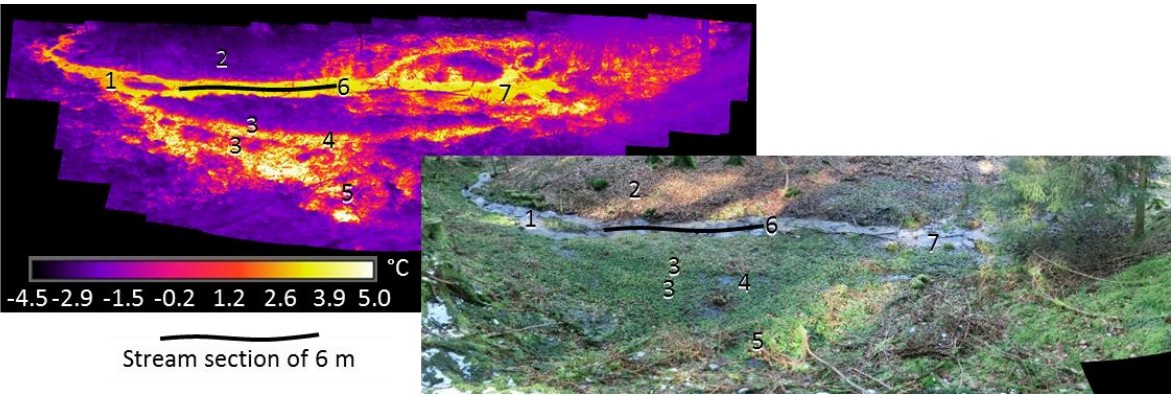

**Figure 1: TIR image and VIS image of a riparian-stream zone. The temperature contrast between the water and the surrounding environment allows us to clearly differentiate between surface-saturated and dry areas in the TIR image. The numbers indicate identical locations in the TIR and VIS images and relate to dry areas (2), stream water (1,4,6), points of supposed groundwater exfiltration (5,7, warmer water temperatures), and locations in which surface saturation is clearly visible in the TIR image but not in the VIS image (3, area above 6).**

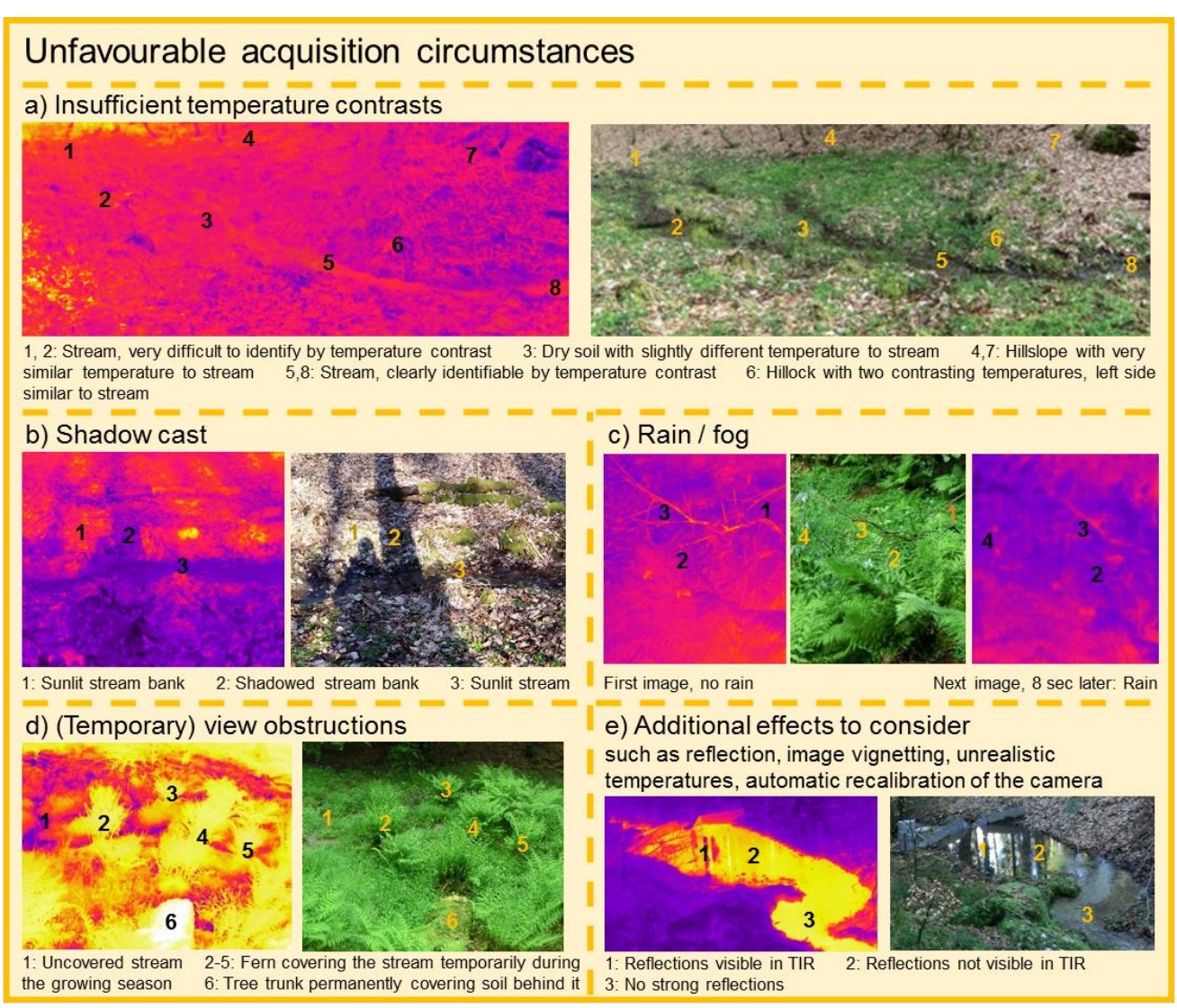

**Figure 2: Example images showing how unfavourable image acquisition circumstances influence the usability of TIR imagery for the identification of surface saturation. The numbers indicate identical locations in the TIR and VIS images.**

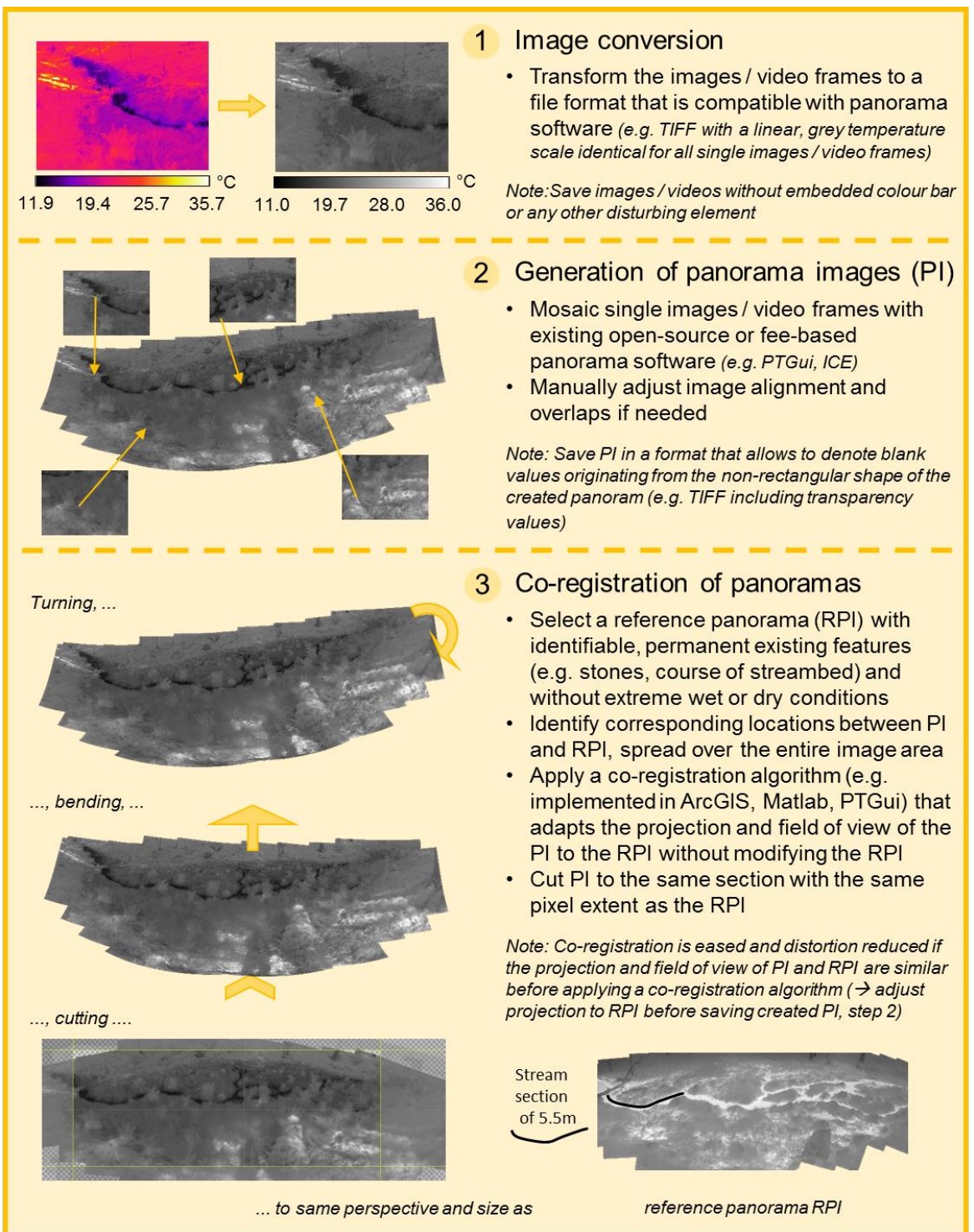

**1  Image conversion**

- Transform the images / video frames to a file format that is compatible with panorama software *(e.g. TIFF with a linear, grey temperature scale identical for all single images / video frames)*

*Note: Save images / videos without embedded colour bar or any other disturbing element*

°C
11.9  19.4  25.7  35.7

°C
11.0  19.7  28.0  36.0

**2  Generation of panorama images (PI)**

- Mosaic single images / video frames with existing open-source or fee-based panorama software *(e.g. PTGui, ICE)*
- Manually adjust image alignment and overlaps if needed

*Note: Save PI in a format that allows to denote blank values originating from the non-rectangular shape of the created panoram (e.g. TIFF including transparency values)*

**3  Co-registration of panoramas**

- Select a reference panorama (RPI) with identifiable, permanent existing features (e.g. stones, course of streambed) and without extreme wet or dry conditions
- Identify corresponding locations between PI and RPI, spread over the entire image area
- Apply a co-registration algorithm (e.g. implemented in ArcGIS, Matlab, PTGui) that adapts the projection and field of view of the PI to the RPI without modifying the RPI
- Cut PI to the same section with the same pixel extent as the RPI

*Note: Co-registration is eased and distortion reduced if the projection and field of view of PI and RPI are similar before applying a co-registration algorithm (→ adjust projection to RPI before saving created PI, step 2)*

*Turning, ...*

*..., bending, ...*

*..., cutting ....*

Stream section of 5.5m

*... to same perspective and size as*          *reference panorama RPI*

**Figure 3: Workflow for processing single TIR images / video frames to co-registered panoramic images.**

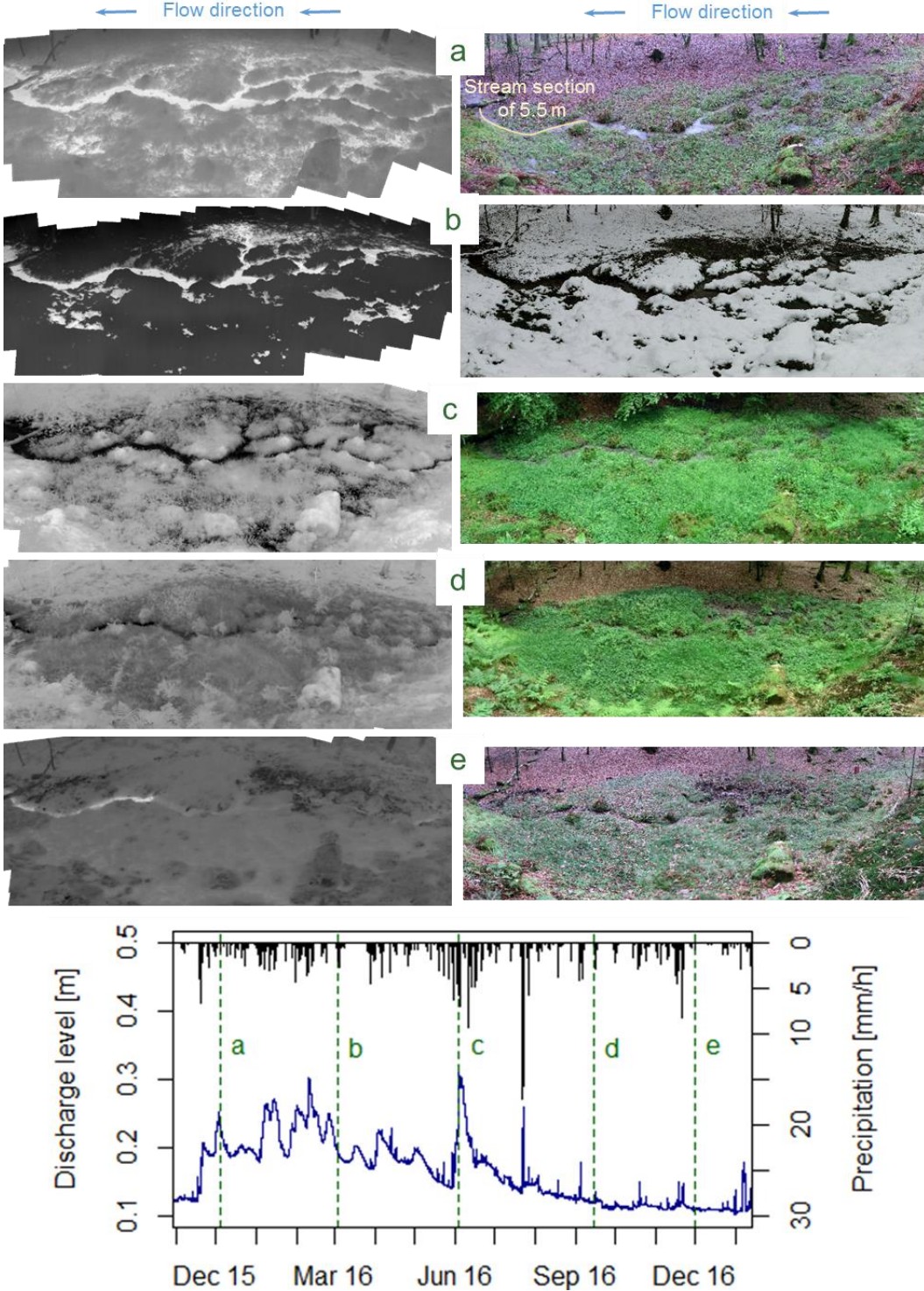

**Figure 4: Time lapse TIR and VIS panoramas showing the variation of surface saturation patterns with varying discharge levels under diverse seasonal conditions.**

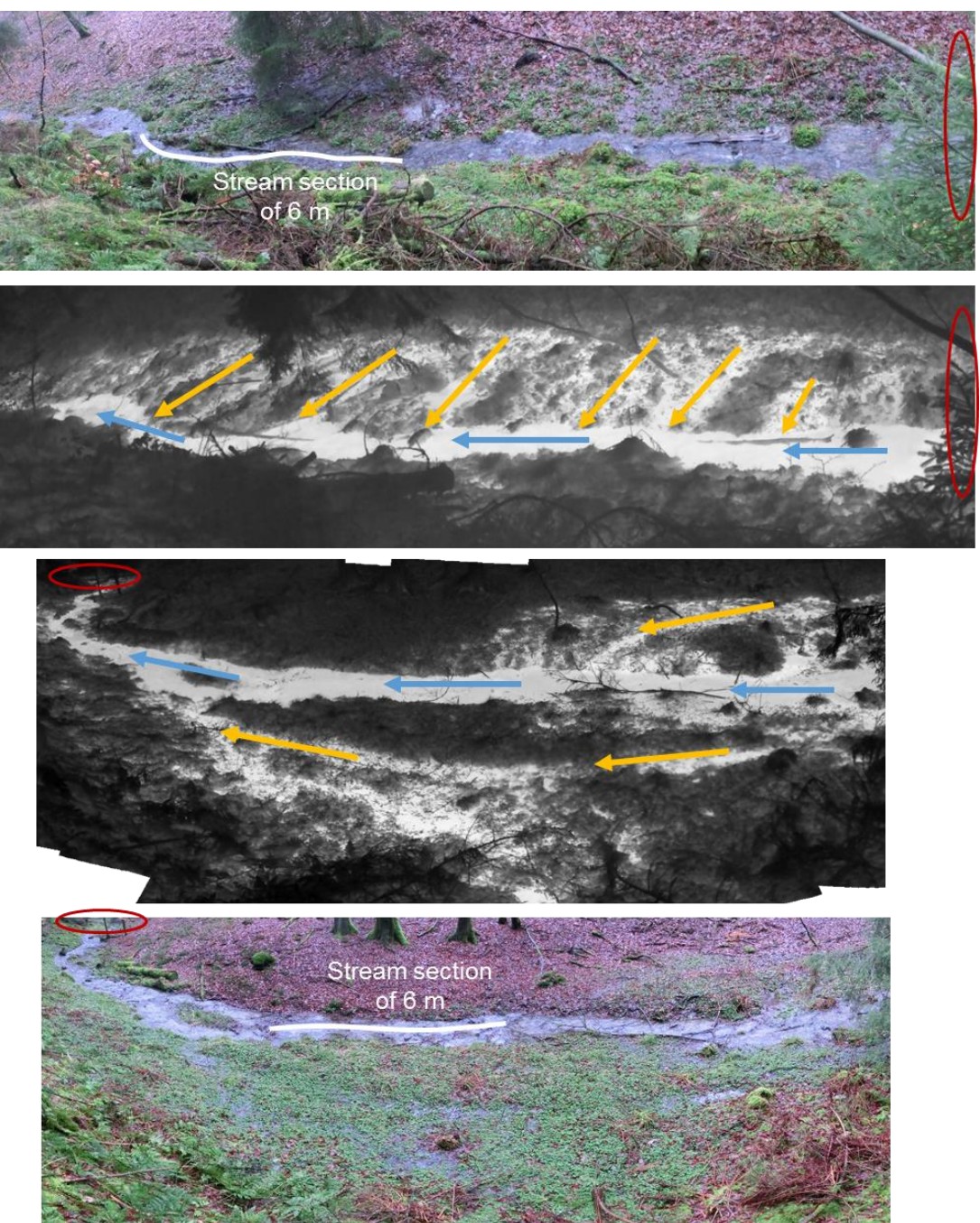

**Figure 5: Comparison of different types of surface saturation patterns. The yellow arrows indicate the orientation of the saturated areas towards the stream (blue arrows = flow direction). The perpendicular direction (top) is likely caused by exfiltrating groundwater connecting to the stream, the parallel direction (bottom) by a parallel flow of the stream expanding into the riparian zone. The red ovals indicate where the two panorama images connect.**

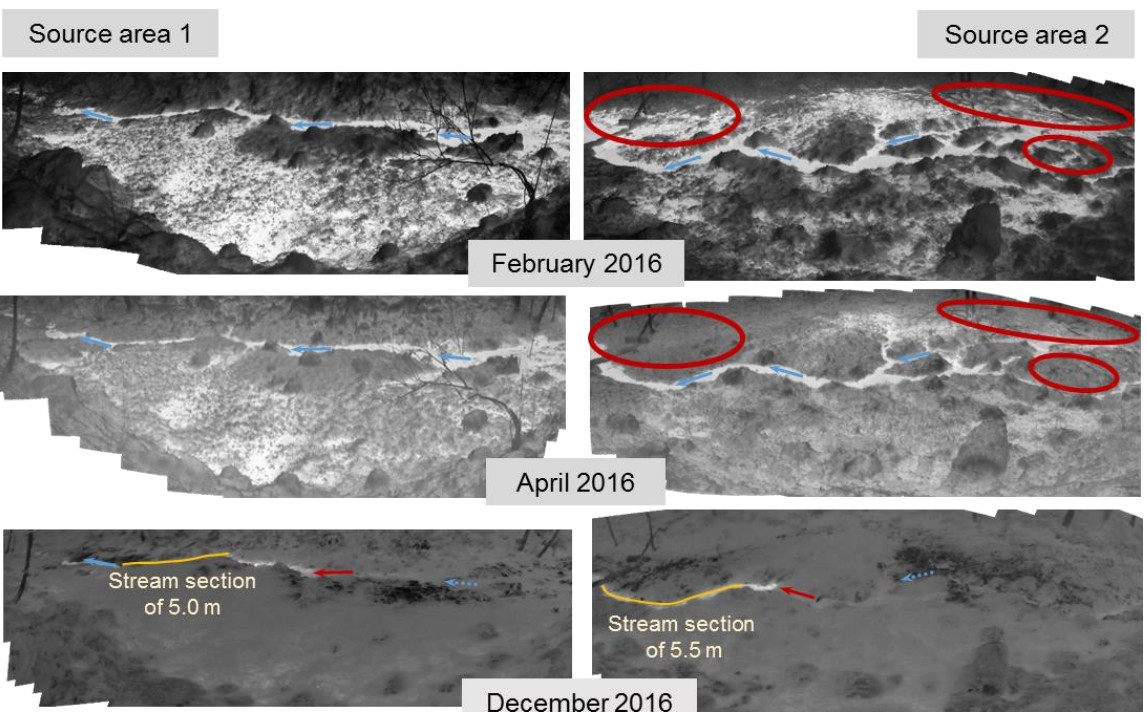

Figure 6: Transition of two source areas (left vs. right) from very wet (top) to very dry conditions (bottom). Surface saturation in source area 1 (left) barely changed between February and April 2016, whereas source area 2 is clearly drier at some locations (red ovals) in April 2016. In December 2016, both source areas were completely dry on each side of the stream (blue arrows = flow direction) and the stream started further downstream (red arrows).

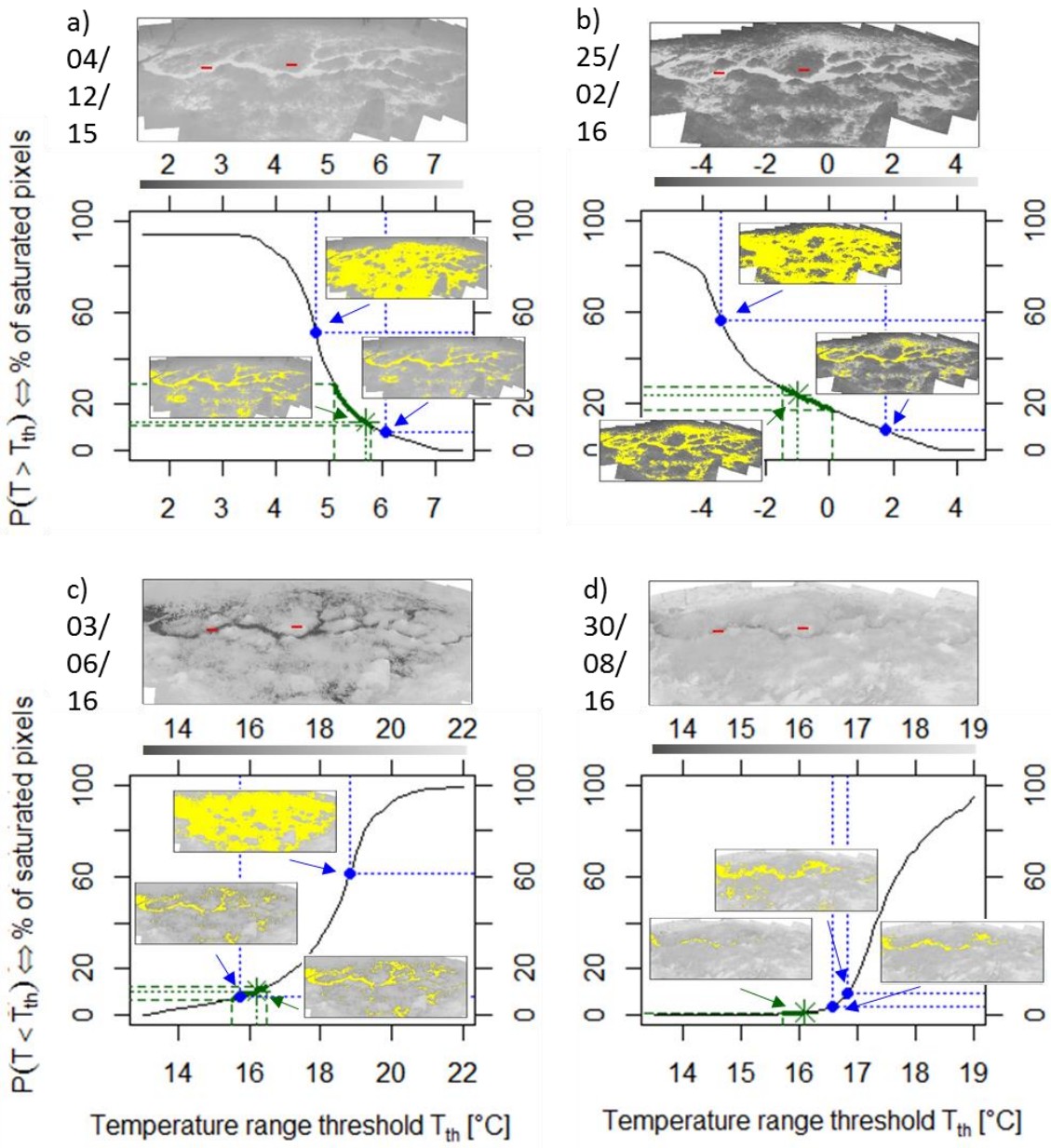

**Figure 7: Example TIR images with their cumulative saturation curves showing the percentage of pixels that have a higher (a, b) or lower (c, d) temperature than the temperature range threshold T$_{th}$ and are thus defined as saturated (marked as yellow pixels in the inset TIR images). The green asterisks mark the temperature ranges that were manually chosen as optimum following a visual assessment of the images. Green dashed lines define the uncertainty of the optimum temperature ranges. The red rectangles in the TIR images depict the masks used for the identification of temperature ranges from a constantly wet (left) and constantly dry (right) area. The respective temperature ranges and saturation percentages are marked in blue. For reference for the spatial dimension of the images, we refer the reader to the indicated stream section in Fig. 3 or Fig. 4.**

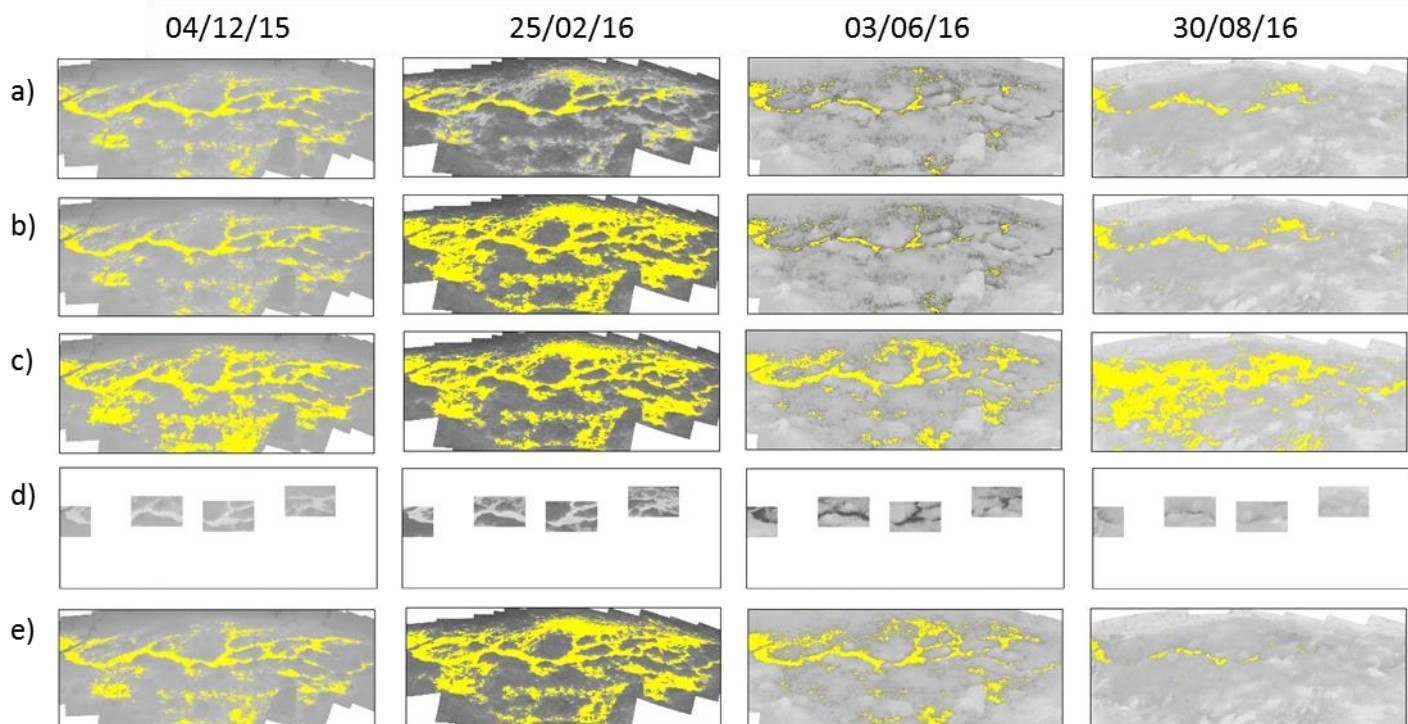

**Figure 8: Comparison of saturation maps (yellow = saturation) generated with a region growing process whose seeds and stopping criteria were automatically constrained to a) bimodal distributions derived from the HBSA applied to the entire image, b) bimodal distributions derived from the HBSA where the selection of bimodal image subsections was constrained to image-specific manual predefinitions of temperature ranges of saturation, c) bimodal distributions derived from pre-selected parts of the image (shown in d) that include clearly wet and dry areas. The saturation maps generated with manually selected temperature ranges based on visual assessment (cf. Fig. 7, green asterisk) are shown for comparison (e). For reference for the spatial dimension of the images, we refer the reader to the indicated stream section in Fig. 3 or Fig. 4.**