# Peer review of "Technical note: Mapping surface saturation dynamics with thermal infrared imagery"

_Hydrology and Earth System Sciences, 2018_

## Referee Comment (RC1) · Anonymous Referee #1 · 5 Aug 2018

The study assesses the practicability of applying thermal infrared (TIR) imagery for mapping surface saturation dynamics. The experimental work was based on an 18-month field campaign, where the authors tried to try to outline under which conditions the method works best and what problems may occur.

General comments

I found the topic very interesting and I really enjoyed reading the paper. Although I do not feel qualified to judge the entire process of acquisition and post-processing of the images since it does not fall within my field of expertise (thus I remind the editor decision to the comments of another more expert reviewer) I see a very high potential for this technique especially its application to larger scales. For this reason, my main comment/suggestion is to put an effort to expand the description of the potential trans-

fer of this technique to larger spatial scales (for instance by using air-born space-born instruments) and the associated difficulties/simplifications/implications this transfer entails.

Technical comment

The paper is really well written and clear. It is also well structure and organized so I do not have specific comments except the suggestions of inclusion of scale bars in the pictures that help the reader to understand the spatial dimension of the figures otherwise too difficult.

Based on that my suggestion is the acceptance of the paper after minor revisions.

---

## Referee Comment (RC2) · Anonymous Referee #2 · 16 Aug 2018

Review of Glaser et al "Mapping Surface Saturation dynamics with thermal infrared imagery"

This technical not describes the opportunities, methodological considerations and challenges of applying thermal infrared imagery to map surface saturation. This technique shows great promise to understand the spatiotemporal dynamics of surface saturation and the hydrological processes that induce or are a result of these dynamics. Overall the manuscript is well written and articulates the challenges and opportunities well and is appropriate as a technical note for HESS. I would recommend publication with minor revisions with the main comments and technical edits provided below.

Main comments:

[Figure]

1. Overall the manuscript conveys a lot of information, but I struggle with the overall organization. Technical notes obviously are not full research articles, but I would still expect a similar format. Intro/Methodology/Results/Discussion/Conclusions. In this work the methodology, results, and discussion seem to overlap in some cases. There is no specific results section, so the findings are not clear before a discussion section begins rather abruptly. I would suggest the following organization:

1. Introduction 2. Methodology

-in fundamental principles it would be good to see the full equation for how to relate what is seen with TIR to absolute temperature. Will help in the communication of the challenges of this method and why for example emissivity and environmental conditions are important.

-in image acquisition if the various challenges could have their own headings. Ie. Weather conditions, view obstruction, view angle... etc..

-"4. Building saturation maps" is still a lot of methodology. Could it be incorporated in this section?

3. Results/Application examples

4. Discussion

5. Conclusions

As is the combination of methodology, results and discussion throughout makes the article feel muddled and at time repetitive even though the information is all very relevant.

2. Generalize. Portions of the article are very specific to the software and camera that were selected for this study. To make this more relevant to a wider audience certain sections could be removed or be made to be more generalized. Ie( Page 5 line33-34, Page 6 line 15-24, and page 7 line 10-19).

3. The influence of difference of surface emissivity's were only very briefly mentioned.

In TIR, depending on what is in the scene, the differences in emissivity's can be important to the reported temperatures- will have implications for absolute temperature values and gradients across the image. I would expect more of a discussion at least so, even if this article doesn't do it, others can incorporate these important corrections in their own work. An example for this in another paper can be found in: Aubry‐Wake, Caroline, et al. "Measuring glacier surface temperatures with ground‐based thermal infrared imaging." Geophysical Research Letters 42.20 (2015): 8489-8497.

Specific comments:

Throughout: please use an oxford comma

Page 2. Line 6-10. Unclear sentence structure

Page 2 Line 16. "up to now" -> to date

Page 2 Line 28-35. Paragraph is muddled. Please improve structure

Page 3 Line 1-2: sentence is awkward

Page 3 Line 13: "Yet," remove

Page 3 Line 18: "allow to obtain an areal picture"- please rewrite as this is awkward

Page 3 Line 29-30: please define surface saturation more clearly on its own as this is critical to the entire paper.

Page 4 Line 3-6. Long complex sentence. Breakup. "or" -> "of"

Page 4 Line 7: "as expressed from" -> relative to

Page 5 Line 14: clarify how one can still observe ground temperatures even if there is vegetation.

Page 6 Line 5. Define what image vignetting is in this situation and why it is a challenge

Page 6 Line 29-34: discussion of consistent temperature scale is redundant

[Figure]

Page 7 Line 21-25. Georectification of terrestrial photos has also been extensively worked on by Corrpio and Harer and should be cited here as well as examples:

Härer, S., M. Bernhardt, J. G. Corripio, and K. Schulz. 2013. "PRACTISE – Photo Rectification And ClassificaTIon SoftwarE (V.1.0)." Geoscientific Model Development 6 (3): 837–848. doi:10.5194/gmd-6-837-2013.

Corripio, J. G. 2004. "Snow Surface Albedo Estimation Using Terrestrial Photography." International Journal of Remote Sensing 25 (24): 5705–5729. doi:10.1080/01431160410001709002.

Page 8 Line 2-4. Can the usability of an image be related to any metrics that would be helpful for fieldwork planning? This would be very useful information from a practical fieldwork perspective- help improve fieldwork efficiency.

Page 9 Line 28-29: please clarify as I'm unclear what the 90% means.

Page 12 Line 21. What does the "(non)-" add to this statement. Confusing as is.

Figure 4. The b scene with snow makes me wonder about how much the snow on the ground combined with the low camera angle is obscuring saturated area. Perhaps discuss this as a challenge in the article.

Figure 7. Please use upper case on first characters of axis labels and put percentage into a unit. "percentage of saturated pixel" is unclear to me. Is this some sort of a cumulative distribution?

---

## Referee Comment (RC3) · Anonymous Referee #3 · 20 Aug 2018

HESS manuscript HESS-2018-334 presents a technical note describing the use of thermal infrared imagery to map surface hydrological saturation. While similar techniques have been previously applied in a limited sense for mapping surface saturation, the manuscript is novel in that it attempts to review and develop best-practice processes for TIR-based saturation mapping. The manuscript is general well-written and thorough and contains a range of good advice. It will therefore be of interest to a broad cross-section of HESS's readership. I have nonetheless included a range of relatively minor comments/suggestions that should be addressed or clarified. Provided the authors are able to make these revisions, I would support its publication in HESS.

—

General comments

- Although the article is generally well written and flows very nicely, some sentences and uses of grammar might appear a bit 'clunky' to native English speakers. This is not to say that the quality of the English isn't already very good, but it might nonetheless be worth passing it to a native English speaker for a quick check.

- The 'Building saturation maps' section is quite long and brings up image processing techniques that are already well-established in the remote sensing literature. It can therefore be shortened. It would also benefit from more consistent referencing to existing image processing/ classification/segmentation literature.

- Although I appreciate that it was not the purpose of the MS, it would still be nice to see some validation of the saturation maps. Do you have any 'squishy boot' data that you can present to validate this data? Or soil moisture data or similar? If not, it would nonetheless be nice to add a section (maybe a paragraph of text) detailing a) the importance of thoroughly validating the TIR data and b) potential validation methods.

—

Abstract

- Consider adding 1-2 sentences at the start of abstract explaining surface saturation dynamics and their importance

P1 L18: Unclear what you mean by 'intuitive character'. Can you rephrase?

P1 L19: No need for comma after 'methodological principles'.

—

Intro

P2 L1: Change 'albeit' to 'despite'.

P2 L9: Change location of 'only': 'Hydrometric measurements ONLY have the potential to monitor. . .'

P2 L18-19: This sentence is factually incorrect - there are numerous examples of research using drones equipped with near infrared cameras to do things like NDVIs. Please delete/re-think this statement.

—

Acquisition of TIR images for mapping surface saturation patterns

P4, L20-22: In my experience, it's less the similarity between air and water temperature that makes identification of waterbodies difficult to detect; it's more when the ground surface temperature and water surface temperature are the same. This means that the 'detectability' of surface water using TIR is often a function of time of day (ie. during summer, water will be cooler than land in the morning, but a similar temperature in the late afternoon/early evening). I know that you've alluded to this in lines 22-26, but it might be nice to clarify this point.

P4 L30: Is 'sun memory effects' an accepted term? Consider rephrasing.

P5 L2: Can you elaborate on why cloudy conditions are better? In my experience, TIR seems to produce better data when there are moderate amounts of cloud (not too clear, not too cloudy, eg. high level cirrus clouds), but persistent low level stratus-type cloud cover can cause reflections that are pernicious in TIR imagery.

P6 L1-3: I believe that the explanation for these false 'negative' values is that during clear sky conditions (if the water is sufficiently still), the water acts like a mirror and reflects the clear sky. However, because the emissivity value at the camera will be set to that of water (0.97) and not of air/sky, the resulting temperature value reported by the camera is incorrect. You get similar effects when filming very reflective surface such as aluminium using TIR cameras.

P6 L10: Change 'shades' to 'shadows'.

P6 L29 – P7 L9 and Figure 3, stage 1: I do not like the proposed technique for colour transformation prior to panorama creation. Personally, I feel that these 'colour transformation' type techniques are sub-optimal, and by converting the images to a simple 8 bit raster, you may lose contrast in the dataset that could be important.

A better way to do it would be to convert the radiometric TIR images to floating-point TIFF files (which preserve 'raw' temperature data), and then create the panorama using these files. This can be done with both of the cameras used in this study (FLIR b425 and T640). Furthermore, if working with videos, it is better to acquire them as .SEQ or .FCF files (essentially sequences of radiometric TIR images) rather than 'conventional' video formats (eg. MP4 files), as these a) preserve temperature data and b) are uncompressed. In my experience, the resulting panoramas are of better quality and have the added benefit of preserving 'real' temperature data.

I do not have a problem with the method you have used, per-se, for the purposes of this manuscript. However, you should not promote this method as the 'best practice' technique for converting TIR images. I would therefore simplify this section of text and Figure 3 to simply say something like 'convert TIR images to a file format that can be used by panorama software'.

P6 L13-20: Similar to the above, the findings here are specific to the panorama software used, and are not necessarily 'best practice'. I would therefore simplify this text, and just talk about the mosaicking process in general terms, rather than talking about the pros/cons of different software packages and using video vs. using still images.

—

Application examples

P8 L15-16: How can you be sure that this is groundwater exfiltration? Could it not just be runoff occurring from a terrace above the stream?

—

Building saturation maps

- This section is quite long, and a lot of the advantages/disadvantages of the image classification/segmentation approaches are common to all types of single-band imagery (not just TIR data). These approaches are thus well established in the remote sensing literature (see histogram thresholding, segmentation using k-means classification, etc etc), and you probably don't need to go into such detail. You could therefore probably shorten this section by around 50% and just preserve the key findings. It might also be nice to include more references to image processing/segmentation/ classification from the remote sensing literature.

—

Discussion

- It would be nice to see some discussion of future work. For example, it would be fairly simple to combine TIR, VIS and NIR data to create multispectral images, thus allowing for advanced image classification procedures to better map connectivity patterns. This would help further improve the quality saturation maps by combining the advantages of these three approaches.

- Furthermore, you could also talk about using new computer vision techniques (eg. deep learning) to improve classification and thus generation of saturation maps from TIR imagery.

—

Figures

Figure 3: See my comments about 'best practice' techniques for converting TIR images to formats that can be used with panorama software. Steps 1 and 2 should be revised to reflect these comments.

Figure 4(e): There is something interesting going on in this image, where the surface water temperature appears to change on the left hand side (ie. much warmer than on the right, as evidences by the light/white colour). Do you know what might be causing

this?

Figure 5. It would be nice to see the visible images that accompany these TIR photos to aid interpretation

---

## Author Comment (AC1) · 23 Aug 2018

Response to Anonymous Referee #1

The study assesses the practicability of applying thermal infrared (TIR) imagery for mapping surface saturation dynamics. The experimental work was based on an 18-month field campaign, where the authors tried to try to outline under which conditions the method works best and what problems may occur.

General comments

I found the topic very interesting and I really enjoyed reading the paper. Although I do not feel qualified to judge the entire process of acquisition and post-processing of the images since it does not fall within my field of expertise (thus I remind the editor decision to the comments of another more expert reviewer) I see a very high potential for this technique especially its application to larger scales.

We wish to thank the reviewer for the positive feedback and we are pleased that the reviewer is interested in the technique and enjoyed reading the manuscript.

For this reason, my main comment/suggestion is to put an effort to expand the description of the potential transfer of this technique to larger spatial scales (for instance by using air-born space-born instruments) and the associated difficulties/simplifications/implications this transfer entails.

We thank the reviewer for pointing out this. We will put more emphasis on the potential scale transfer and related challenges in the discussion section.

Technical comment

The paper is really well written and clear. It is also well structure and organized so I do not have specific comments except the suggestions of inclusion of scale bars in the pictures that help the reader to understand the spatial dimension of the figures otherwise too difficult.

The images are taken with a low angle and the images are thus spatially distorted. This means that it won't be possible to include scale bars that are valid for all of the image. In order to facilitate the feeling for the spatial dimension we will indicate the length of a distinct stream section within the images instead.

Based on that my suggestion is the acceptance of the paper after minor revisions.

---

## Author Comment (AC2) · 23 Aug 2018

Response to Anonymous Referee #2

This technical not describes the opportunities, methodological considerations and challenges of applying thermal infrared imagery to map surface saturation. This technique shows great promise to understand the spatiotemporal dynamics of surface saturation and the hydrological processes that induce or are a result of these dynamics. Overall the manuscript is well written and articulates the challenges and opportunities well and is appropriate as a technical note for HESS. I would recommend publication with minor revisions with the main comments and technical edits provided below.

We wish to thank the reviewer for the assessment of the manuscript and for the valuable comments and suggested edits.

Main comments:

1. Overall the manuscript conveys a lot of information, but I struggle with the overall organization. Technical notes obviously are not full research articles, but I would still expect a similar format. Intro/Methodology/Results/Discussion/Conclusions. In this work the methodology, results, and discussion seem to overlap in some cases. There is no specific results section, so the findings are not clear before a discussion section begins rather abruptly. I would suggest the following organization:

1. Introduction 2. Methodology
-in fundamental principles it would be good to see the full equation for how to relate what is seen with TIR to absolute temperature. Will help in the communication of the challenges of this method and why for example emissivity and environmental conditions are important.
-in image acquisition if the various challenges could have their own headings. Ie.Weather conditions, view obstruction, view angle… etc..
-"4. Building saturation maps" is still a lot of methodology. Could it be incorporated in this section?

3. Results/Application examples

4. Discussion

5. Conclusions

As is the combination of methodology, results and discussion throughout makes the article feel muddled and at time repetitive even though the information is all very relevant.

We thank the reviewer for the suggestions on the structure, this is an important point.
The current manuscript combines both review and own experimental work on how to use the TIR methodology (see objectives in the introduction). Thus we think that the organization in the 'classical' way is not the best option.
However, we agree that especially section 4 can be clearly improved in terms of structure and can be disentangled. We will carefully recheck the manuscript for overlaps of methodologies, results, and discussion within the different sections and we will revise the structure of the manuscript to some extent. Mainly, we will integrate section 3 'Application examples' into section 2. Section 4 will be subdivided, but be a stand-alone section. We decided to do this since we have one method review and methodological approach for section 2-3 and section 4 each. In other words, the two parts evaluate and review different aspects of the technology. We hope that our planned restructuring will make this clearer.

2. Generalize. Portions of the article are very specific to the software and camera that were selected for this study. To make this more relevant to a wider audience certain sections could be removed or be made to be more generalized. Ie( Page 5 line33-34, Page 6 line 15-24, and page 7 line 10-19).

We agree that some parts are very specific and we will generalize them in the revised manuscript (see also our reply to the comment of Reviewer #3 on P6, L 13-20). Yet, we think that not all parts can be completely generalized (i.e. P5, L33-34), but that these parts are nonetheless interesting for several camera types / software and thus might be useful for a wider audience.

3. The influence of difference of surface emissivity's were only very briefly mentioned. In TIR, depending on what is in the scene, the differences in emissivity's can be important to the reported temperatures- will have implications for absolute temperature values and gradients across the image. I would expect more of a discussion at least so, even if this article doesn't do it, others can incorporate these important corrections in their own work. An example for this in another paper can be found in: Aubryâ˘A˘RWake,Caroline, et al. "Measuring glacier surface temperatures with groundâ˘AR˘ based thermalinfrared imaging." Geophysical Research Letters 42.20 (2015): 8489-8497.

We will give it more emphasis in a revised form of the manuscript.

Specific comments:

We thank the reviewer for the technical comments and suggestions, we will consider them in a revised form of the manuscript.

Throughout: please use an oxford comma
Page 2. Line 6-10. Unclear sentence structure
Page 2 Line 16. "up to now" -> to date
Page 2 Line 28-35. Paragraph is muddled. Please improve structure
Page 3 Line 1-2: sentence is awkward
Page 3 Line 13: "Yet," remove
Page 3 Line 18: "allow to obtain an areal picture"- please rewrite as this is awkward
Page 3 Line 29-30: please define surface saturation more clearly on its own as this is critical to the entire paper.
Page 4 Line 3-6. Long complex sentence. Breakup. "or" -> "of"
Page 4 Line 7: "as expressed from" -> relative to
Page 5 Line 14: clarify how one can still observe ground temperatures even if there is vegetation.
Page 6 Line 5. Define what image vignetting is in this situation and why it is a challenge
Page 6 Line 29-34: discussion of consistent temperature scale is redundant

Page 7 Line 21-25. Georectification of terrestrial photos has also been extensively worked on by Corrpio and Harer and should be cited here as well as examples:

Härer, S., M. Bernhardt, J. G. Corripio, and K. Schulz. 2013. "PRACTISE – Photo Rectification And ClassificaTIon SoftwarE (V.1.0)." Geoscientific Model Development 6 (3): 837–848. doi:10.5194/gmd-6-837-2013.

Corripio, J. G. 2004. "Snow Surface Albedo Estimation Using Terrestrial Photography."International Journal of Remote Sensing 25 (24): 5705–5729.doi:10.1080/01431160410001709002.

Page 8 Line 2-4. Can the usability of an image be related to any metrics that would be helpful for fieldwork planning? This would be very useful information from a practical fieldwork perspective- help improve fieldwork efficiency.

Sadly we cannot relate this to any metric. The usability depends on the combination of various conditions affecting the image quality as mentioned and explained in section 2.2. In addition, it depends on the intended purpose up to which point an image is useable.
Fieldwork efficiency can be improved by avoiding such unfavourable conditions, but it is unfortunately not possible to predict with certainty if the temperature contrast will be good enough. Thus, in our case it was still a third of images that was not useable.

Page 9 Line 28-29: please clarify as I'm unclear what the 90% means.
Page 12 Line 21. What does the "(non)-" add to this statement. Confusing as is.
Figure 4. The b scene with snow makes me wonder about how much the snow on the ground combined with the low camera angle is obscuring saturated area. Perhaps discuss this as a challenge in the article.

Snow can indeed obscure the saturated area. We mentioned snow as a possible view obstruction (P5, L12), but we will add one or two sentences to further discuss the effect of snow. If the amount of snow is low, the saturated areas mainly stay uncovered (due to a warmer temperature of the water

and thus a fast melting). This is the case for Figure 4b. If the snow cover is thicker, saturated areas will be covered and the snow surface has to be interpreted as the new ground surface.

Figure 7. Please use upper case on first characters of axis labels and put percentage into a unit. "percentage of saturated pixel" is unclear to me. Is this some sort of a cumulative distribution?

---

## Author Comment (AC3) · 23 Aug 2018

Response to Anonymous Referee #3

HESS manuscript HESS-2018-334 presents a technical note describing the use of thermal infrared imagery to map surface hydrological saturation. While similar techniques have been previously applied in a limited sense for mapping surface saturation, the manuscript is novel in that it attempts to review and develop best-practice processes for TIR-based saturation mapping. The manuscript is general well-written and thorough and contains a range of good advice. It will therefore be of interest to a broad cross-section of HESS's readership. I have nonetheless included a range of relatively minor comments/suggestions that should be addressed or clarified. Provided the authors are able to make these revisions, I would support its publication in HESS.

We wish to thank the reviewer for the assessment of the manuscript and for the helpful and detailed comments and suggestions.

—
General comments

- Although the article is generally well written and flows very nicely, some sentences and uses of grammar might appear a bit 'clunky' to native English speakers. This is not to say that the quality of the English isn't already very good, but it might nonetheless be worth passing it to a native English speaker for a quick check.

We will use our internal institutional language editing.

- The 'Building saturation maps' section is quite long and brings up image processing techniques that are already well-established in the remote sensing literature. It can therefore be shortened. It would also benefit from more consistent referencing to existing image processing/classification/ segmentation literature.

We agree that the section is rather long and includes techniques that are well known for a reader with expertise in remote sensing. Yet, we think that the given information will be helpful and important for readers with a different background (e.g. field hydrologists). These two aspects will need to be balanced as one purpose of the work is to establish these methods more in the field of experimental hydrology. We will carefully screen the section in order to shorten where possible and we will follow the reviewer's advice and include more references from the remote sensing literature. In addition, we will improve the structure of this section (see also our response to Referee #2 concerning the structure), which should also allow for a remote sensing expert to decide if parts of the section can be skipped for reading.

- Although I appreciate that it was not the purpose of the MS, it would still be nice to see some validation of the saturation maps. Do you have any 'squishy boot' data that you can present to validate this data? Or soil moisture data or similar? If not, it would nonetheless be nice to add a section (maybe a paragraph of text) detailing a) the importance of thoroughly validating the TIR data and b) potential validation methods.

We basically used our observations and experience from the field and the respective VIS images as ground truth / validation. We will include a separate section to explicitly state this in the revised manuscript. In this new section we will also discuss why other data than visual observation are difficult to be used as ground truth (soil moisture shows the saturation in the upper soil layer, not on the ground surface; the squishy boot method also includes water that gets squeezed out of the soil when stepping on it, whereas the non-invasive TIR imagery won't detect this).

—
Abstract

- Consider adding 1-2 sentences at the start of abstract explaining surface saturation dynamics and their importance
P1 L18: Unclear what you mean by 'intuitive character'. Can you rephrase?
P1 L19: No need for comma after 'methodological principles'.

We thank the reviewer for the remarks, we will include them in a revised form of the manuscript.

—
Intro

P2 L1: Change 'albeit' to 'despite'.
P2 L9: Change location of 'only': 'Hydrometric measurements ONLY have the potential to monitor…'
P2 L18-19: This sentence is factually incorrect - there are numerous examples of research using drones equipped with near infrared cameras to do things like NDVIs. Please delete/re-think this statement.
True, reading the sentence as it is, the statement is incorrect and needs to be rephrased. The intention was to refer it only to research using NDVI/NDWI for mapping surface saturation. We are not aware of any study that does this with a drone or ground-based. If the reviewer is, we would be happy if (s)he could provide us a reference for this.

—
Acquisition of TIR images for mapping surface saturation patterns

P4, L20-22: In my experience, it's less the similarity between air and water temperature that makes identification of waterbodies difficult to detect; it's more when the ground surface temperature and water surface temperature are the same. This means that the 'detectability' of surface water using TIR is often a function of time of day (ie. During summer, water will be cooler than land in the morning, but a similar temperature in the late afternoon/early evening). I know that you've alluded to this in lines 22-26, but it might be nice to clarify this point.
P4 L30: Is 'sun memory effects' an accepted term? Consider rephrasing.
P5 L2: Can you elaborate on why cloudy conditions are better? In my experience, TIR seems to produce better data when there are moderate amounts of cloud (not too clear, not too cloudy, eg. high level cirrus clouds), but persistent low level stratus-type cloud cover can cause reflections that are pernicious in TIR imagery.
P6 L1-3: I believe that the explanation for these false 'negative' values is that during clear sky conditions (if the water is sufficiently still), the water acts like a mirror and reflects the clear sky. However, because the emissivity value at the camera will be set to that of water (0.97) and not of air/sky, the resulting temperature value reported by the camera is incorrect. You get similar effects when filming very reflective surface such as aluminium using TIR cameras.
P6 L10: Change 'shades' to 'shadows'.

We agree with the comments above. We will consider them in a revised form of the manuscript, though we might spare some details.

P6 L29 – P7 L9 and Figure 3, stage 1: I do not like the proposed technique for colour transformation prior to panorama creation. Personally, I feel that these 'colour transformation' type techniques are sub-optimal, and by converting the images to a simple 8 bit raster, you may lose contrast in the dataset that could be important.
A better way to do it would be to convert the radiometric TIR images to floating-point TIFF files (which preserve 'raw' temperature data), and then create the panorama using these files. This can be done with both of the cameras used in this study (FLIR b425 and T640). Furthermore, if working with videos, it is better to acquire them as .SEQ or .FCF files (essentially sequences of radiometric TIR images) rather than 'conventional' video formats (eg. MP4 files), as these a) preserve temperature data and b) are uncompressed. In my experience, the resulting panoramas are of better quality and have the added benefit of preserving 'real' temperature data.
I do not have a problem with the method you have used, per-se, for the purposes of this manuscript. However, you should not promote this method as the 'best practice' technique for converting TIR images. I would therefore simplify this section of text and Figure 3 to simply say something like 'convert TIR images to a file format that can be used by panorama software'.

We will carefully recheck the formulations and given information of this section to ensure that it does not read as a 'best practice' but simply as one possible method. We acquired both the images and the videos as radiometric TIR images (thus as .SEQ files for the videos). However, we do not know a standard software that can read these radiometric files for creating a panorama. Therefore, we exported the images in a more standard file format. In order to retain the temperature information we exported the images as grey-scale images, knowing which colour value corresponds to which temperature value. Saving the images as floating-point TIFFs instead did not appear to us to be more advantageous in terms of contrast/detail and practicability than the grey-converted images. This might depend on specific image characteristics and we would be open for further discussions on this.

P6 L13-20: Similar to the above, the findings here are specific to the panorama software used, and are not necessarily 'best practice'. I would therefore simplify this text, and just talk about the mosaicking process in general terms, rather than talking about the pros/cons of different software packages and using video vs. using still images.

Referee #2 pointed out as well that some parts of the manuscript (including this one) should be generalized. We will simplify and shorten this section.

—

Application examples

P8 L15-16: How can you be sure that this is groundwater exfiltration? Could it not just be runoff occurring from a terrace above the stream?

We think the reviewer's suggestion of adding the VIS images to Figure 5 will help to clarify this. There is no 'terrace' above the stream.

[Figure]

—

Building saturation maps

- This section is quite long, and a lot of the advantages/disadvantages of the image classification/segmentation approaches are common to all types of single-band imagery (not just TIR data). These approaches are thus well established in the remote sensing literature (see histogram thresholding, segmentation using k-means classification, etc etc), and you probably don't need to go into such detail. You could therefore probably shorten this section by around 50% and just preserve the key findings. It might also be nice to include more references to image processing/segmentation/classification from the remote sensing literature.

See our reply to the second general comment.

—

Discussion

- It would be nice to see some discussion of future work. For example, it would be fairly simple to combine TIR, VIS and NIR data to create multispectral images, thus allowing for advanced image classification procedures to better map connectivity patterns. This would help further improve the quality saturation maps by combining the advantages of these three approaches.
- Furthermore, you could also talk about using new computer vision techniques (eg. deep learning) to improve classification and thus generation of saturation maps from TIR imagery.

Both are good and interesting points and we will extend the discussion accordingly.

—

Figures

Figure 3: See my comments about 'best practice' techniques for converting TIR images to formats that can be used with panorama software. Steps 1 and 2 should be revised to reflect these comments.

Figure 4(e): There is something interesting going on in this image, where the surface water temperature appears to change on the left hand side (ie. much warmer than on the right, as evidences by the light/white colour). Do you know what might be causing this?

This change in temperature is very likely caused by the fact that the temperature of warm, exfiltrating groundwater (starting point of the stream on the right) is influenced and changed by the temperature of the surroundings. Especially since the water volume/depth is low, this can happen within such a short distance.

Figure 5. It would be nice to see the visible images that accompany these TIR photos to aid interpretation

---

## Author Response (AR1)

**Response to Anonymous Referee #1**

The study assesses the practicability of applying thermal infrared (TIR) imagery for mapping surface saturation dynamics. The experimental work was based on an 18-month field campaign, where the authors tried to try to outline under which conditions the method works best and what problems may occur.

General comments

I found the topic very interesting and I really enjoyed reading the paper. Although I do not feel qualified to judge the entire process of acquisition and post-processing of the images since it does not fall within my field of expertise (thus I remind the editor decision to the comments of another more expert reviewer) I see a very high potential for this technique especially its application to larger scales.

We wish to thank the reviewer for the positive feedback and we are pleased that the reviewer is interested in the technique and enjoyed reading the manuscript.

For this reason, my main comment/suggestion is to put an effort to expand the description of the potential transfer of this technique to larger spatial scales (for instance by using air-born space-born instruments) and the associated difficulties/simplifications/implications this transfer entails.

We thank the reviewer for pointing out this. We emphasised the potential scale transfer and related challenges by adding a paragraph on this in the discussion section (P13, L4-14).

Technical comment

The paper is really well written and clear. It is also well structure and organized so I do not have specific comments except the suggestions of inclusion of scale bars in the pictures that help the reader to understand the spatial dimension of the figures otherwise too difficult.

The images are taken with a low angle and the images are thus spatially distorted. This means that it is not possible to include scale bars that are valid for all of the image. In order to facilitate the perception for the spatial dimension we indicated the length of a distinct stream section within the images instead (with the exception of Fig. 2, since this would have overloaded the images)

Based on that my suggestion is the acceptance of the paper after minor revisions.

**Response to Anonymous Referee #2**

This technical not describes the opportunities, methodological considerations and challenges of applying thermal infrared imagery to map surface saturation. This technique shows great promise to understand the spatiotemporal dynamics of surface saturation and the hydrological processes that induce or are a result of these dynamics. Overall the manuscript is well written and articulates the challenges and opportunities well and is appropriate as a technical note for HESS. I would recommend publication with minor revisions with the main comments and technical edits provided below.

We wish to thank the reviewer for the assessment of the manuscript and for the valuable comments and suggested edits. Main comments:

1. Overall the manuscript conveys a lot of information, but I struggle with the overall organization. Technical notes obviously are not full research articles, but I would still expect a similar format. Intro/Methodology/Results/Discussion/Conclusions. In this work the methodology, results, and discussion seem to overlap in some cases. There is no specific results section, so the findings are not clear before a discussion section begins rather abruptly. I would suggest the following organization:

1. Introduction 2. Methodology
-in fundamental principles it would be good to see the full equation for how to relate what is seen with TIR to absolute temperature. Will help in the communication of the challenges of this method and why for example emissivity and environmental conditions are important.

We included the equation for the Stefan Boltzmann law and for the radiometric corrections (P3, L30 - P4, L9).

-in image acquisition if the various challenges could have their own headings. Ie.Weather conditions, view obstruction, view angle… etc..

We added three sub-headings (Impact of weather conditions, Camera position, and Measurement artefacts during image acquisition)

-"4. Building saturation maps" is still a lot of methodology. Could it be incorporated in this section?

3. Results/Application examples

4. Discussion

5. Conclusions

As is the combination of methodology, results and discussion throughout makes the article feel muddled and at time repetitive even though the information is all very relevant.

We thank the reviewer for the suggestions on the structure. It is very important to us that the structure of the manuscript is easy to follow and comprehensible.
Yet, as the manuscript combines both review and own experimental work on how to use the TIR methodology (see objectives in the introduction) the organization in the 'classical' way is in our opinion not the best option.
We agree with the reviewer that the structure was a bit muddled and iterated the structure of the manuscript and improved various sections:
1. We integrated section 3 'Application examples' of the previous manuscript into section 2 (now 2.4).
2. We disentangled section 3 'Quantification of saturation through pixel classification' (former section 4 'Building saturation maps') and subdivided it into two parts, 'Methods for generating binary saturation maps' and 'Comparison of methods for generating binary saturation maps for TIR images'.

The reason behind this revision of the structure is that the manuscript contains two parts that evaluate different aspects of the TIR technology. The first part (now section two) deals with the identification of saturation. The second part (section 3 in the new manuscript) assesses approaches on how to generate binary saturation maps from the images. Each of the two sections consists of the methodological aspects but also on a review of current literature. We round the paper with a general discussion on both key sections and the conclusions. This structure is also explicitly outlined at the end of the Introduction of the new version of the manuscript (P3, L21-25).

2. Generalize. Portions of the article are very specific to the software and camera that were selected for this study. To make this more relevant to a wider audience certain sections could be removed or be made to be more generalized. Ie( Page 5 line33-34, Page 6 line 15-24, and page 7 line 10-19).

We agree that some parts are very specific and we generalized especially the descriptions in section 2.3 (see also our reply to the comment of Reviewer #3 on P6, L 13-20). Yet, we did not remove all parts that could not be completely generalized (i.e. P5, L33-34, now P7, L3-5), since we think that the information contained in these parts is also of interest for users employing different camera types / software.

3. The influence of difference of surface emissivity's were only very briefly mentioned. In TIR, depending on what is in the scene, the differences in emissivity's can be important to the reported temperatures- will have implications for absolute temperature values and gradients across the image. I would expect more of a discussion at least so, even if this article doesn't do it, others can incorporate these important corrections in their own work. An example for this in another paper can be found in: AubryˇAˇRWake,Caroline, et al. "Measuring glacier surface temperatures with groundˆAˇR based thermalinfrared imaging." Geophysical Research Letters 42.20 (2015): 8489-8497.

We put more emphasis on it in the revised manuscript (P6, L34 - P7, L3).

Specific comments:

Throughout: please use an oxford comma We now consistently use the oxford comma
Page 2. Line 6-10. Unclear sentence structure We reformulated the sentences.
Page 2 Line 16. "up to now" -> to date Done
Page 2 Line 28-35. Paragraph is muddled. Please improve structure We rearranged the paragraph
Page 3 Line 1-2: sentence is awkward We rewrote the sentence.
Page 3 Line 13: "Yet," remove Done
Page 3 Line 18: "allow to obtain an areal picture"- please rewrite as this is awkward Done
Page 3 Line 29-30: please define surface saturation more clearly on its own as this is critical to the entire paper. We removed the first half of the sentence to make the definition more stand alone.
Page 4 Line 3-6. Long complex sentence. Breakup. "or" -> "of" Done
Page 4 Line 7: "as expressed from" -> relative to Thanks for that suggestion
Page 5 Line 14: clarify how one can still observe ground temperatures even if there is vegetation. We clarified that this depends on the density of the vegetation cover.
Page 6 Line 5. Define what image vignetting is in this situation and why it is a challenge We added two sentences to address this.
Page 6 Line 29-34: discussion of consistent temperature scale is redundant We removed it

Page 7 Line 21-25. Georectification of terrestrial photos has also been extensively worked on by Corrpio and Harer and should be cited here as well as examples:

Härer, S., M. Bernhardt, J. G. Corripio, and K. Schulz. 2013. "PRACTISE – Photo Rectification And ClassificaTIon SoftwarE (V.1.0)." Geoscientific Model Development 6 (3): 837–848. doi:10.5194/gmd-6-837-2013.

Corripio, J. G. 2004. "Snow Surface Albedo Estimation Using Terrestrial Photography." International Journal of Remote Sensing 25 (24): 5705–5729. doi:10.1080/01431160410001709002.

We added the two references in the manuscript.

5   Page 8 Line 2-4. Can the usability of an image be related to any metrics that would be helpful for fieldwork planning? This would be very useful information from a practical fieldwork perspective- help improve fieldwork efficiency.

Sadly, we cannot relate this to any metric. The applicability depends on the combination of various conditions affecting the image quality as mentioned and explained in section 2.2. In addition, it depends on the intended purpose up to which point an 10 image can be used.
Fieldwork efficiency can be improved by avoiding such unfavourable conditions, but it is not possible to predict with certainty if the temperature contrast will be good enough.

Page 9 Line 28-29: please clarify as I'm unclear what the 90% means. We clarified the formulation (P11, L11-12).

Page 12 Line 21. What does the "(non)-" add to this statement. Confusing as is.
We deleted "(non-)permanent"

Figure 4. The b scene with snow makes me wonder about how much the snow on the ground combined with the low camera 20 angle is obscuring saturated area. Perhaps discuss this as a challenge in the article.
Snow can indeed obscure the saturated area. We mentioned snow as a possible view obstruction and added two sentences to further discuss the effect of snow in the revised manuscript (P6, L12-14). If the amount of snow is low, the saturated areas mainly stay uncovered (due to a warmer temperature of the water and thus a fast melting). This is the case for Figure 4b. If the snow cover is thicker, saturated areas will be covered and the snow surface has to be interpreted as the new ground surface.

Figure 7. Please use upper case on first characters of axis labels and put percentage into a unit. "percentage of saturated pixel" is unclear to me. Is this some sort of a cumulative distribution?
We changed the axis labels as demanded. We also clarified in the labels that '% of saturated pixels' is equivalent to all pixels having a higher/lower temperature than a temperature range threshold. So yes, the plot shows the cumulative saturation 30 distribution.

**Response to Anonymous Referee #3**

HESS manuscript HESS-2018-334 presents a technical note describing the use of thermal infrared imagery to map surface hydrological saturation. While similar techniques have been previously applied in a limited sense for mapping surface saturation, the manuscript is novel in that it attempts to review and develop best-practice processes for TIR-based saturation mapping. The manuscript is general well-written and thorough and contains a range of good advice. It will therefore be of interest to a broad cross-section of HESS's readership. I have nonetheless included a range of relatively minor comments/suggestions that should be addressed or clarified. Provided the authors are able to make these revisions, I would support its publication in HESS.

10   We wish to thank the reviewer for the assessment of the manuscript and for the helpful and detailed comments and suggestions.

—

General comments

- Although the article is generally well written and flows very nicely, some sentences and uses of grammar might appear a bit
15   'clunky' to native English speakers. This is not to say that the quality of the English isn't already very good, but it might nonetheless be worth passing it to a native English speaker for a quick check.

We used our internal institutional language editing.

20   - The 'Building saturation maps' section is quite long and brings up image processing techniques that are already well-established in the remote sensing literature. It can therefore be shortened. It would also benefit from more consistent referencing to existing image processing/classification/ segmentation literature.

We agree that the section is rather long and includes techniques that are well-established in remote sensing literature. We
25   should of course not present the obvious or repeat findings from other fields. Yet, we think that the given information is necessary to understand the approaches, especially for readers with a different background (e.g. field hydrologists). These two aspects need to be balanced as one purpose of the work is to establish these methods more in the field of experimental hydrology. Eventually, we did not considerably shorten the section. Instead, we improved the structure of this section by dividing it into two subsections: '3.1 Methods for generating binary saturation maps' and '3.2 Comparison of methods for
30   generating binary saturation maps for TIR images' (see also our response to Referee #2 concerning the structure). We think that this will allow for a remote sensing expert to decide if especially the first part of the section can be skipped for reading. In addition, we followed the reviewer's advice and included more references from the remote sensing literature in the first part of the section to demonstrate the common use of these approaches at other scales.

35   - Although I appreciate that it was not the purpose of the MS, it would still be nice to see some validation of the saturation maps. Do you have any 'squishy boot' data that you can present to validate this data? Or soil moisture data or similar? If not, it would nonetheless be nice to add a section (maybe a paragraph of text) detailing a) the importance of thoroughly validating the TIR data and b) potential validation methods.

40   We basically used our observations from the field and the respective VIS images as ground truth / validation. We included a separate paragraph to explicitly state this in the 'Fundamental principles' section. In this paragraph we also discuss why other data than visual observation are difficult to be used as ground truth (P5, L7-14). We also added a sentence on the ground truth data used for comparing the different pixel classification approaches (P10, L26-27).

45   —

Abstract

- Consider adding 1-2 sentences at the start of abstract explaining surface saturation dynamics and their importance Done
P1 L18: Unclear what you mean by 'intuitive character'. Can you rephrase? Done.

P1 L19: No need for comma after 'methodological principles'. We removed it

—

Intro

P2 L1: Change 'albeit' to 'despite'. Done

P2 L9: Change location of 'only': 'Hydrometric measurements ONLY have the potential to monitor…'

We rephrased the sentence to make it shorter and clearer (cf. comment of Referee #2) and don't use 'only' anymore.

P2 L18-19: This sentence is factually incorrect - there are numerous examples of research using drones equipped with near infrared cameras to do things like NDVIs. Please delete/re-think this statement.

Indeed, this was not well formulated and we changed it. Our intention was to refer only to research using NDVI/NDWI for mapping surface saturation. We are not aware of any study that does this with a drone or ground-based and we rephrased the sentence accordingly. If the reviewer is aware of a study using drones to map surface saturation with NDVI/NDWI, we would be happy if (s)he could provide us a reference for this.

—

Acquisition of TIR images for mapping surface saturation patterns

P4, L20-22: In my experience, it's less the similarity between air and water temperature that makes identification of waterbodies difficult to detect; it's more when the ground surface temperature and water surface temperature are the same. This means that the 'detectability' of surface water using TIR is often a function of time of day (ie. During summer, water will be cooler than land in the morning, but a similar temperature in the late afternoon/early evening). I know that you've alluded to this in lines 22-26, but it might be nice to clarify this point.

We agree and rephrased the sentence to clarify the point on the temperature similarities. The link to the time of day is mentioned at the end of the section (P5, L32-P6, L2).

P4 L30: Is 'sun memory effects' an accepted term? Consider rephrasing. We replaced it.

P5 L2: Can you elaborate on why cloudy conditions are better? In my experience, TIR seems to produce better data when there are moderate amounts of cloud (not too clear, not too cloudy, eg. high level cirrus clouds), but persistent low level stratus-type cloud cover can cause reflections that are pernicious in TIR imagery.

We agree with your experience. However, we think that this may be overly complex to predict. Generally, we advise cloudy conditions in order to avoid the effect of sun. This is specified in the revised manuscript (P6, L2-3).

P6 L1-3: I believe that the explanation for these false 'negative' values is that during clear sky conditions (if the water is sufficiently still), the water acts like a mirror and reflects the clear sky. However, because the emissivity value at the camera will be set to that of water (0.97) and not of air/sky, the resulting temperature value reported by the camera is incorrect. You get similar effects when filming very reflective surface such as aluminium using TIR cameras.

We added one sentence about the possible explanation (P7, L6-8).

P6 L10: Change 'shades' to 'shadows'. Done

P6 L29 – P7 L9 and Figure 3, stage 1: I do not like the proposed technique for colour transformation prior to panorama creation. Personally, I feel that these 'colour transformation' type techniques are sub-optimal, and by converting the images to a simple 8 bit raster, you may lose contrast in the dataset that could be important.

A better way to do it would be to convert the radiometric TIR images to floating-point TIFF files (which preserve 'raw' temperature data), and then create the panorama using these files. This can be done with both of the cameras used in this study (FLIR b425 and T640). Furthermore, if working with videos, it is better to acquire them as .SEQ or .FCF files (essentially sequences of radiometric TIR images) rather than 'conventional' video formats (eg. MP4 files), as these a) preserve
5  temperature data and b) are uncompressed. In my experience, the resulting panoramas are of better quality and have the added benefit of preserving 'real' temperature data.

I do not have a problem with the method you have used, per-se, for the purposes of this manuscript. However, you should not promote this method as the 'best practice' technique for converting TIR images. I would therefore simplify this section of text and Figure 3 to simply say something like 'convert TIR images to a file format that can be used by panorama software'.

We carefully checked the formulations and given information of this section to ensure that it does not read as a 'best practice' but rather as one possible approach. We acquired the images and the videos as radiometric TIR images (thus as .SEQ files for the videos). However, we do not know a standard software that can read these radiometric files for creating a panorama. Therefore, we exported the images and videos in a more standard file format (JPEG, .wmv). We exported the images as grey-
15  scale images / video frames, knowing which colour value corresponds to which temperature value, in order to retain the temperature information. Relying on floating-point TIFFs instead did not appear to us to be more advantageous in terms of contrast/detail and practicability than the grey-converted images. This might depend on specific image characteristics and we are open for further discussions and exchange on this.

20  P6 L13-20: Similar to the above, the findings here are specific to the panorama software used, and are not necessarily 'best practice'. I would therefore simplify this text, and just talk about the mosaicking process in general terms, rather than talking about the pros/cons of different software packages and using video vs. using still images.

Referee #2 pointed out as well that some parts of the manuscript (including this one) should be generalized. We simplified and
25  shortened this section to account for this.

—

Application examples

P8 L15-16: How can you be sure that this is groundwater exfiltration? Could it not just be runoff occurring from a terrace
30  above the stream?

We think the reviewer's suggestion of adding the VIS images to Figure 5 will help to clarify this. There is no 'terrace' above the stream.

[Figure]

—

Building saturation maps

- This section is quite long, and a lot of the advantages/disadvantages of the image classification/segmentation approaches are
40  common to all types of single-band imagery (not just TIR data). These approaches are thus well established in the remote

sensing literature (see histogram thresholding, segmentation using k-means classification, etc etc), and you probably don't need to go into such detail. You could therefore probably shorten this section by around 50% and just preserve the key findings. It might also be nice to include more references to image processing/segmentation/classification from the remote sensing literature.

5   See our reply to the second general comment.

—
Discussion

- It would be nice to see some discussion of future work. For example, it would be fairly simple to combine TIR, VIS and NIR
10  data to create multispectral images, thus allowing for advanced image classification procedures to better map connectivity patterns. This would help further improve the quality saturation maps by combining the advantages of these three approaches.

- Furthermore, you could also talk about using new computer vision techniques (eg. deep learning) to improve classification and thus generation of saturation maps from TIR imagery.
15
Both are good and interesting points and we took them up at the end of the discussion. (P14, L2-6).

—
Figures

20  Figure 3: See my comments about 'best practice' techniques for converting TIR images to formats that can be used with panorama software. Steps 1 and 2 should be revised to reflect these comments.

We revised step 1. Step 2 is identical to the original version, since we think that this step is valid for all panorama software, independently of how the images were converted.
25
Figure 4(e): There is something interesting going on in this image, where the surface water temperature appears to change on the left hand side (ie. much warmer than on the right, as evidences by the light/white colour). Do you know what might be causing this?

30  This change in temperature is very likely caused by the fact that the temperature of warm, exfiltrating groundwater (starting point of the stream on the right) is influenced and changed by the temperature of the surroundings. Since the water volume/depth is low, this can happen within a very short distance

Figure 5. It would be nice to see the visible images that accompany these TIR photos to aid interpretation
35
We added the visual images for both areas.

[revised manuscript text omitted]

**Unfavourable acquisition circumstances**

**a) Insufficient temperature contrasts**

1, 2: stream, very difficult to identify by temperature contrast    3: dry soil with slightly different temperature than stream    4,7: hillslope with very similar temperature as stream    5,8: stream, clearly identifiable by temperature contrast    6: hillock with two contrasting temperatures, left side similar to stream

**b) Shadow cast**

1: sunlit stream bank    2: shadowed stream bank    3: sunlit stream

**c) Rain / fog**

First image, no rain                Next image, 8 sec later: Rain

**d) (Temporary) view obstructions**

1: uncovered stream    2-5: fern covering the stream temporary during the growing season    6: tree trunk permanently covering soil behind

**e) Additional effects to consider**
such as reflection, image vignetting, unrealistic temperatures, automatic recalibration of the camera

1: reflections visible in TIR    2: reflections not visible in TIR    3: no strong reflections

[revised manuscript text omitted]